

# **Discrete k-nearest neighbor resampling for simulating multisite precipitation occurrence and adaption to climate change**

: Discrete KNNR for Multisite Occurrence (DKMO version1.0) - model development

Keywords: daily precipitation, discrete, k-nearest neighbor, Markov chain, multisite, occurrence

Taesam Lee[1] and Vijay P. Singh[2]

[1] Department of Civil Engineering, ERI, Gyeongsang National University,

501 Jinju-daero, Jinju, Gyeongnam, South Korea, 660-701

[2] Department of Biological and Agricultural Engineering & Zachry Department of Civil Engineering, Texas A&M University, 321 Scoates Hall, College Station, Texas, United States, 77843

Corresponding Author :

Taesam Lee, Ph.D.
Gyeongsang National University, Dept. of Civil Engineering
Tel)+82-55-772-1797, Fax)+82-55-772-1799
Email) tae3lee@gnu.ac.kr

# **Abstract**

Stochastic weather simulation models are commonly employed in water resources management, agricultural applications, forest management, transportation management, and recreational activities. Stochastic simulation of multisite precipitation occurrence is a challenge because of its intermittent characteristics as well as spatial and temporal cross-correlation. This study proposes a novel simulation method for multisite precipitation occurrence employing a nonparametric technique, the discrete version of the k-nearest neighbor resampling (KNNR), and coupling it with Genetic Algorithm (GA). Its modification for the study of climatic change adaptation is also tested. The datasets simulated from both the DKNNR model and an existing traditional model were evaluated using a number of statistics, such as occurrence and transition probabilities as well as temporal and spatial cross-correlations. Results showed that the proposed DKNNR model with GA simulated multisite precipitation occurrence preserved the lagged crosscorrelation between sites while the existing conventional model was not able to reproduce lagged crosscorrelation between stations, so long stochastic simulation was required. Also, the GA mixing process provided a number of new patterns that were different from observations, which was not feasible with the sole DKNNR model. When climate change was considered, the model performed satisfactorily, but further improvement is required to more accurately simulate specific variations of the occurrence probability.

## 1. Introduction

Stochastic simulation of weather variables has been employed for water resources management, hydrological design, agricultural irrigation, forest management, transportation planning and evacuation, recreation activities, filling-in missing historical data, simulating data, extending observed records, and simulating different weather conditions. Stochastic simulation models play a key role in producing weather sequences, while preserving the statistical characteristics of observed data. A number of stochastic weather simulation models have been developed using parametric and nonparametric approaches (Lee, 2017; Lee et al., 2012; Wilby et al., 2003; Wilks, 1999; Wilks and Wilby, 1999).

Parametric approaches simulate statistical characteristics of observed weather data with a set of parameters that are determined by fitting (Jeong et al., 2012; Lee, 2016; Zheng and Katz, 2008), whereas in nonparametric approaches, historical analogs with current conditions are searched, following the weather simulation data (Buishand and Brandsma, 2001; Lee et al., 2012). Combinations of parametric and nonparametric approaches have also been proposed (Apipattanavis et al., 2007; Frost et al., 2011).

Among weather variables, precipitation possesses intermittency and zero values between precipitation events, which make it difficult to properly reproduce the events (Beersma and Buishand, 2003; Hughes et al., 1999; Katz and Zheng, 1999). To overcome the problem of intermittency and zero values, precipitation is simulated separately from other variables. The main method for reproducing intermittency has been the multiplication of precipitation occurrence and an amount as $Z=X \cdot Y$, where $X$ is the occurrence (binary as either 0 or 1) and $Y$ is the amount (Jeong et al., 2013; Lee and Park, 2017; Todorovic and Woolhiser, 1975). The spatial and temporal

dependence in the occurrence and amount of precipitation introduces further complexity in
multisite simulation.
Wilks (1998) presented a multisite simulation model for the occurrence process (i.e. $X$) using
the standard normal variable that is spatially dependent, representing the relation between the
occurrence variable and the standard normal variable with simulation data. Originally, the
occurrence of precipitation had been simulated with a discrete Markov Chain (MC) model (Katz,
1977). Compared to the MC model that requires a significant number of parameters for generating
multisite occurrence, the multisite occurrence model proposed by Wilks (1998) transforms the
standard normal variate and simulates the sequence with multivariate normal distribution, and then
back-transforms the multivariate normal sequence to the original domain. The model is able to
reproduce the contemporaneous multisite dependence structure and lagged dependence only for
the same site but it requires a complex simulation process to estimate parameters for each site and
is unable to preserve lagged dependence between sites.  Also, a recent improvement has also been
made, but the weakness of the model in Wilks (1998) was not significantly improved (Evin et al.,
2018; Mehrotra et al., 2006; Srikanthan and Pegram, 2009).
Lee et al. (2010a) proposed a nonparametric-based stochastic simulation model for
hydrometeorological variables. Their model overcame the shortcomings of a previous
nonparametric simulation model (Lall and Sharma, 1996), called k-nearest neighbor resampling
(KNNR) but the simulated data do not produce patterns different from those of the observed data
(Brandsma and Buishand, 1998; Mehrotra et al., 2006; St-Hilaire et al., 2012). In addition to
KNNR, Lee et al. (2010a) used a meta-heuristic Genetic Algorithm (GA) that led to the
reproduction of similar populations by mixing the simulated datasets. Note that the reproduction
procedure of the GA allows to generate new patterns that are similar to observed patterns, but a
small number of totally new patterns are simulated from the mutation procedure of the GA.
While KNNR is employed to find historical analogues of multisite occurrence similar to the
current status of a simulation series, GA is applied to use its skill to generate a new descendant
from the historical parent chosen with the KNNR. In this procedure, the multisite occurrence of
precipitation can be simulated while preserving spatial and temporal correlations. Meta-heuristic
techniques, such as GA, have been popularly employed in a number of hydrometeorological
applications (Chau, 2017; Fotovatikhah et al., 2018; Taormina et al., 2015; Wang et al., 2013).
Although a number of variants of KNNR-GA have been applied (Lee et al., 2012; Lee and Park,
2017), none of them can simulate multisite occurrence of precipitation whose characteristics are
binary and temporally and spatially related.
Therefore, this study proposes a stochastic simulation method for multisite occurrence of
precipitation with the KNNR-GA based nonparametric approach that (1) simulates multisite
occurrence with a simple and direct procedure without parameterization of all the required
occurrence probabilities; and (2) reproduces the complex temporal and spatial correlation between
stations as well as the basic occurrence probabilities. The proposed nonparametric model is
compared with the popular model proposed by Wilks (1998). Even though the multisite occurrence
data generated from the Wilks model preserves various statistical characteristics of the observed
data well, significant underestimation of lagged cross-correlation still exists. Furthermore, the
relation between standard normal variable and occurrence variable relies on long stochastic
simulation.
The paper is organized as follows. The next section presents the mathematical background
of existing multisite occurrence modeling and section discusses the modeling procedure. The
study area and data are reported in section 4. The model application is presented in section 5.
Results of the proposed model are discussed in section 6, and summary and conclusions are
presented in section 7.

## 2. Background

### 2.1.  Single site occurrence modeling

Let $X_t^s$ represent the occurrence of daily precipitation for a location $s$ ($s=1,\ldots, S$) on day $t$
($t=1,\ldots, n$; $n$ is the number observed days) and let $X_t^s$ be either zero for dry day or one for wet day.
The first order Markov chain model for $X_t^s$ is defined with the assumption that the occurrence
probability of a wet day is fully defined by the previous day as

$$\Pr\{X_t^s = 1 \mid X_{t-1}^s = 0\} = p_{01}^s \tag{1}$$

$$\Pr\{X_t^s = 1 \mid X_{t-1}^s = 1\} = p_{11}^s \tag{2}$$

Also $p_{00}^s = 1 - p_{01}^s$ and $p_{10}^s = 1 - p_{11}^s$ , since the summation of zero and one should be unity
with the same previous condition. This consists of a transition probability matrix (TPM) as

$$TPM^s = \begin{bmatrix} p_{00}^s & p_{01}^s \\ p_{10}^s & p_{11}^s \end{bmatrix} = \begin{bmatrix} 1 - p_{01}^s & p_{01}^s \\ 1 - p_{11}^s & p_{11}^s \end{bmatrix} \tag{3}$$

The marginal distributions of TPM (i.e. $p_0$ and $p_1$) can be expressed with TPM and its condition of
$p_0 + p_1 = 1$ as:

$$p_0^s = \frac{p_{01}^s}{1 + p_{01}^s - p_{11}^s} \tag{4}$$

$$p_1^s = \frac{1 - p_{11}^s}{1 + p_{01}^s - p_{11}^s} \qquad (5)$$

Note that $p_1$ represents the probability of precipitation occurrence for a day, while $p_0$ does non-
occurrence. The lag-1 autocorrelation of precipitation occurrence is the combination of transition
probabilities as:
$$\rho_1(s,s) = p_{11}^s - p_{01}^s \qquad (6)$$

The simulation can be done by comparing TPM with a uniform random number ($u_t^s$) as
$$X_t^s = \begin{cases} 1 & \text{if } u_t^s \le p_{i1}^s \\ 0 & \text{otherwise} \end{cases} \qquad (7)$$

where $p_{i1}^s$ is the selected probability from TPM regarding the previous condition $i$ (i.e. either 0 or
1). Wilks (1998) suggested a different method using a standard normal random number $w_t^s \sim N[0,1]$
as
$$X_t^s = \begin{cases} 1 & \text{if } w_t^s \le \Phi^{-1}(p_{i1}^s) \\ 0 & \text{otherwise} \end{cases} \qquad (8)$$

where $\Phi^{-1}$ indicates the inverse of the standard normal cumulative function $\Phi$.

### 2.2. Multisite occurrence modeling

Wilks (1998) suggested a multisite occurrence model using a standard normal random
number (here, denoted as MONR) that is spatially dependent but serially independent. The
correlation of the standard normal variate for a site pair of $q$ and $s$ can be expressed as:
$$\tau(q,s) = corr[w_t^q, w_t^s] \qquad (9)$$

Also, the correlation of the original occurrence variate is

$$\rho(q,s) = corr[X_t^q, X_t^s] \qquad (10)$$

Once the correlation of the standard normal variate is known, the simulation of multisite
precipitation occurrence is straightforward. Multivariate standard normal distribution  is used with
a parameter set of $[\mathbf{0}, \mathbf{T}]$ where $\mathbf{0}$ is the zero vector ($S$x1) and $\mathbf{T}$ is the correlation matrix with the
elements of $\tau(q,s)$ for $q \in \{1,...,S\}$ and $s \in \{1,...,S\}$.
Since direct estimation of $\tau(q,s)$ is not feasible, a simulation technique is used to estimate
$\tau(q,s)$ from $\rho(q,s)$. A long sequence of the occurrences is simulated with different values of
$\tau(q,s)$ and its corresponding correlation of the original domain $\rho(q,s)$ is estimated with the
simulated long sequence by the inverse standard normal cumulative function (i.e. $\Phi^{-1}$). A curve
between $\tau(q,s)$ and $\rho(q,s)$ is derived from this long simulation with the MONR model and is
employed for parameter estimation for a real application.

## 3. DKNNR


### 3.1. DKNNR modeling procedure


In the current study, a novel multisite simulation model for discrete occurrence of precipitation
variable with k-nearest neighbor resampling (KNNR) technique (Lall and Sharma, 1996; Lee
and Ouarda, 2011; Lee et al., 2017) for a discrete case (denoted as Discrete KNNR; DKNNR)
is proposed by combining a mixture mechanism with Genetic Algorithm (GA). Provided the
number of nearest neighbors, $k$, is known, the discrete k-nearest neighbor resampling with
genetic algorithm is done as follows:
(1) Estimate the distance between the current (i.e. time index: c) multisite occurrence

$X_c^s$ and the observed multisite occurrence $x_i^s$. Here, the distance is measured for

$i=1,\ldots, n$-1 as

$$D_i = \sum_{s=1}^{S} \left| X_c^s - x_i^s \right| \tag{11}$$

(2) Arrange the estimated distances from step (1) in ascending order, select the first $k$

distances (i.e., the smallest $k$ values), and reserve the time indices of the smallest $k$

distances.

(3) Randomly select one of the stored $k$ time indices with the weighting probability

given by

$$w_m = \frac{1/m}{\sum_{j=1}^{k} 1/j} \ , \qquad m = 1,\ldots, k \tag{12}$$

(4) Assume the selected time index from step (3) as $p$. Note that there are a number of

values that have the same distance as the selected $D_p$, since $D_p$ is a natural number

between 0 and $S$. For example, if $S=2$ and $X_c^1=0$ and $X_c^2=1$, the two sequences have

the same $D=1$ as $[x_i^1=0$ and $x_i^2=0]$ and $[x_i^1=1$ and $x_i^2=1]$. In this case, a random

selection procedure is required to take into account the cases with the same quantity.

One particular time index is randomly selected with equal probabilities among the

time indices of the same distances. Note that instead of the random selection, one

can always use the first one. In such a case, only one historical combination of

multisite occurrences will be selected.

(5) Assign the binary vector of the proceeding index of the selected time as

$\mathbf{x}_{p+1} = [x_{p+1}^s]_{s \in \{1,S\}}$. Here, $p$ is the finally selected time index from step (4).

(6) Execute the following steps for GA mixing if GA mixing is subjectively selected.

Otherwise, skip this step.

(6-1) Reproduction: Select one additional time index using steps (1) through (4) and

denote this index as $p^*$. Obtain the corresponding precipitation occurrence

values, $\mathbf{x}_{p^*+1} = [x_{p^*+1}^s]_{s \in \{1,\ldots,S\}}$. The subsequent two GA operators employ the two

selected vectors, $\mathbf{x}_{p+1}$ and $\mathbf{x}_{p^*+1}$. This reproduction process is a mating process

by finding another individual that has characteristics similar to those of the

current one $\mathbf{x}_{p+1}$. With this procedure, a vector similar to the current vector will

be mated and will produce a new descendant.

(6-2) Crossover: Replace each element $x_{p+1}^s$ with $x_{p^*+1}^s$ at probability $P_{cr}$, i.e.,

$$X_{c+1}^s = \begin{cases} x_{p^*+1}^s & \text{if } \varepsilon < P_{cr} \\ x_{p+1}^s & \text{otherwise} \end{cases} \tag{13}$$

where $\varepsilon$ is a uniform random number between 0 and 1. From this crossover, a

new occurrence vector whose elements are similar to the historical ones is generated.

(6-3) Mutation: Replace each element (i.e., each station, $s=1,\ldots, S$) with one selected

from all the observations of this element for $i=1,\ldots, n$ with probability $P_m$, i.e.,

$$X_{c+1}^s = \begin{cases} x_{\xi+1}^s & \text{if } \varepsilon < P_m \\ x_{p+1}^s & \text{otherwise} \end{cases} \tag{14}$$

where $x^s_{\xi+1}$ is selected from $[x^s_i]_{i\in\{1,\ldots,n\}}$ with equal probability for $i=1,\ldots,n$

and $\varepsilon$ is a uniform random number between 0 and 1. This mutation procedure

allows to generate a multisite occurrence combination that is totally different

from the historical records. Without this procedure, multisite occurrences

always similar to historical combinations are generated, which is not feasible

for a simulation purpose.

(7) Repeat steps (1)-(6) until the required data are generated.

The selection of the number of nearest neighbors ($k$) has been investigated by Lall and
Sharma (1996) and Lee and Ouarda (2011). A simple selection method was applied in the current
study as $k = \sqrt{n}$. For hydrometeorological stochastic simulation, this heuristic approach of the $k$
selection has been employed  (Lall and Sharma, 1996; Lee and Ouarda, 2012; Lee et al., 2010b;
Prairie et al., 2006; Rajagopalan and Lall, 1999).  One can use generalized cross-validation (GCV)
as shown in Sharma and Lall (1996) and Lee and Ouarda 2011 by treating this simulation as a
prediction problem. However, the current multisite occurrence simulation does not necessarily
require an accurate value prediction and not much difference in simulation using the simple
heuristic approach has been reported. Also, this heuristic approach of the $k$ selection has been
popularly employed for hydrometeorological stochastic simulations (Lall and Sharma, 1996; Lee
and Ouarda, 2012; Lee et al., 2010b; Prairie et al., 2006; Rajagopalan and Lall, 1999).
In Appendix A, an example of the DKNNR simulation procedure is explained in detail.

## 3.2. Adaptation to climate change

The capability of model to take climate change into account is critical. For example, the
marginal distributions and transition probabilities in Eqs. (5) and (3) can change in future climate
scenarios. It is known that nonparametric simulation models have a difficulty to adapt to climate
change, since the models employ in general the current observation sequences. However, the
proposed model in the current study possesses the capability to adapt to the variations of
probabilities by tuning the crossover and mutation probabilities in $P_{cr}$ (13) and $P_m$ (14), adding the
condition when applied.

For example, the probability of $P_{11}$ can be increased with the cross-over probability $P_{cr}$ by

adding the condition to increase the probability of $P_{11}$ as:
$$X_{c+1}^s = \begin{cases} x_{p*+1}^s & \text{if } \varepsilon < P_{cr} \text{ \& } x_{p*+1}^s = 1 \text{ \& } X_c^s = 1 \\ x_{p+1}^s & \text{otherwise} \end{cases} \tag{15}$$

It is obviously possible to increase the probability of $P_1$ by removing the condition of $X_c^s = 1$.

In addition, further adjustment can be made with the mutation process in Eq. (14) as

$$X_{c+1}^s = \begin{cases} x_{\xi+1}^s & \text{if } \varepsilon < P_m \text{ and } x_{\xi+1}^s = 1 \\ x_{p+1}^s & \text{otherwise} \end{cases} \tag{16}$$

This adjustment of adding the condition $x_{\xi+1}^s = 1$ can increase the marginal distribution as much as
$P_m \times P_1$. This has been tested in a case study.

## 4. Study area and data description

For testing the occurrence model, 12 weather stations were selected from Yeongnam province
which is located in the southeastern part of South Korea, as shown in Figure 1. Information on
longitude and latitude (fourth and fifth columns) as well as order index and the identification
number (first and second columns) of these stations operated by Korea Meteorological
Administration with the area name (third column) is shown in Table 1. The employed precipitation
dataset presents strong seasonality, since this area is dry from late fall to early autumn and humid
and rainy during the remaining seasons, especially in summer. The employed stations are not far
from each other, at most 100 km apart, and not much high mountains are located in the current
study area. Therefore, this region can be considered as a homogeneous region (Lee et al., 2007).

Figure 1 illustrates the locations of the selected weather stations. All the stations are inside

Yeongnam province which consists of two different regions as north and south Gyeongsang as
well as the self-governing cities of Busan, Daegu, and Ulsan. Most of the Yeongnam region is
drained to Nakdong River. To validate the proposed model appropriately, test sites must be highly
correlated with each other as well as have significant temporal relation. The stations inside the
Yeongnam area cover one of the most important watersheds, the Nakdong River basin, where the
Nakdong River passes through the entire basin and its hydrological assessments for agriculture
and climate change have a particular value in flood control and water resources management such
as floods and droughts.

It is important to analyze the impact of weather conditions for planning agricultural

operations and water resources management, especially during the summer season, because around
50-60 percent of the annual precipitation occurs during the summer season from June to September.
The length of daily precipitation data record ranges from 1976 to 2015 and the summer season
record was employed, since a large number of rainy days occur during summer and it is important
to preserve these characteristics. Also, the whole year dataset was tested and other seasons were
further applied but the correlation coefficient was relatively high and its correlation matrix
estimated was not a positive semi-definite matrix for the MONR model.

## 5. Application

To analyze the performance of the proposed DKNNR model, the occurrence of precipitation was simulated. The DKNNR simulation was compared with that of the MONR model. For each model, 100 series of daily occurrence with the same record length were simulated. The key statistics of observed data and each generated series, such as transition probabilities ($P_{11}$, $P_{01}$, and $P_1$) and cross-correlation (see Eq.(10)), were determined. The MONR model underestimated the lag-1 cross-correlation, as indicated by Wilks (1998). In the current study, this statistic was analyzed, since a synoptic scale weather system often results in lagged cross-correlation for daily precipitation data (Wilks, 1998). It was formulated as

$$\rho_1(q,s) = corr[X_{t-1}^q, X_t^s] \tag{17}$$

Statistics from 100 generated series were evaluated by the root mean square error (RMSE) expressed as:

$$RMSE = \left( \frac{1}{N} \sum_{m=1}^{N} (\Gamma_m^G - \Gamma^h)^2 \right)^{1/2} \tag{18}$$

where $N$ is the number of series (here 100), $\Gamma_m^G$ is the statistic estimated from the $m^{\text{th}}$ generated series, while $\Gamma^h$ is the statistic for the observed data. Note that lower RMSE indicates better performance, represented by the summarized error of a given statistic of generated series from the statistic of the observed data.

The 100 simulated statistic values were illustrated with boxplots to show their variability as shown in Figure 5 - Figure 7. The box of boxplot represents the interquartile range (IQR) ranging from 25 percentile to 75 percentile. The whiskers extend to up and down 1.5×IQR. Data beyond

the whiskers (1.5×IQR) are indicated by a plus sign (+). The horizontal line inside the box
represents the median of the data. The statistics of the observed data are denoted by a cross (x).
The closer a cross is to the horizontal line inside the box, the better the simulated data from a model
reproduces the statistical characteristics of the observed data.

## 6. Results

### 6.1.   GA mixing and its probability selection

The roles of crossover probability $P_{cr}$ (Eq. (13)) and mutation probability $P_m$ (Eq.(14)) were studied
by Lee et al. (2010b). In the current study, we further tested by selecting an appropriate parameter
set of these two parameters with the simulated data from the DKNNR model and the record length
of 100,000. RMSE (Eq. (18)) of the three transition and limiting probabilities ($P_{11}$, $P_{01}$, and $P_1$)
between the simulated data and the observed was used, since those probabilities are key statistics
that the simulated data must match the observed data and no parameterization of these probabilities
was made for the current DKNNR model. Results are shown in Figure 2 and Figure 3 for $P_{cr}$ and
$P_m$, respectively. For $P_{cr}$ in Figure 2, the probability of 0.02 shows the smallest RMSE in all
transition and limiting probabilities. The RMSE of $P_m$ in Figure 3 shows a slight fluctuation along
with $P_m$. However, all three probabilities ($P_{11}$, $P_{01}$, and $P_1$) have relatively small RMSEs in $P_m$
=0.003. Therefore, the parameter set 0.02 and 0.003 was chosen for $P_{cr}$ and $P_m$, respectively, and
employed in the current study. We also tested the simulation without the GA mixing procedure
(results not shown). The results showed that no better result could be found from the simulation
without GA mixing. The necessity of the GA mixing is further discussed in the following.
We further tested and discuss why the GA mixing is necessary in the proposed DKNNR
model as follows. For example, assume that three weather stations are considered and observed
data only has the occurrence cases of 000, 001,011,010, 011,100,111 among $2^3=8$ possible cases.
In other words, no patterns for 110 and 101 is found in the observed data. Note that 0 is dry day
and 1 is rainy (or wet) day. The KNNR is a resampling process in that the simulation data is
resampled from observations. Therefore, no new patterns such as 110 and 101 can be found in the
simulated data.

This can be problematic for the simulation purpose in that one of the major simulation

purposes is to simulate sequences that might possibly happen in the future. The wet (1) or dry (0)
for multisite precipitation occurrence is decided by the spatial distribution of a precipitation
weather system. A humid air mass can be distributed randomly, relying on wind velocity and
direction as well as the surrounding air pressure. In general, any combinations of wet and dry
stations can be possible, especially when the simulation continues infinitely. Therefore, the
patterns of simulated data must be allowed to have any possible combinations, here 4096 even if
it has not been observed from the historical records. Also, its probability to have this new pattern
must not be high, since it has not been observed in the historical records and this can be taken into
account by low probability of the crossover and mutation.

This drawback of the KNNR model frequently happens in multisite occurrence as the

number of stations increases. Note that the number of patterns increases as $2^n$ where $n$ is the number
of stations. If $n=12$, then 4096 cases must be observed. However, among 4096 cases, observed
cases are limited, since the number of data is limited. The GA process can mix two candidate
patterns to produce new patterns. For example, in the three station case, a new pattern 101 can be
produced from two observed occurrence candidates of 001 and 100 by the crossover of the first
value of 001 to the first value of 100 (i.e. 001 $\rightarrow$101), which is not in the observed data.

Note that the data employed in the case study are 40 years and 122 days (summer months)

in each year. The total number of the observed data is 4880 and the number of possible cases is
4096. We checked the number of possible cases that were not found in the observed data. The
result shows that 3379 cases were not observed at all for the entire cases as shown in Figure 4.

We further investigated the number of new patterns that were generated with the

probabilities $P_{cr}$=0.02, $P_m$=0.001 by the proposed GA mixing. The generated data for 100
sequences from DKNNR with the GA mixing shows that the number 3379 was reduced to 1200,
which is not in the dataset among the 4096 possible patterns. Therefore, more than 2000 new
patterns were simulated with the GA mixing process. The KNNR model without the GA mixing
did not produce any new patterns in the 100 sequences with the same length of the historical data.

## 6.2.   Occurrence and transition probabilities

The data simulated from the proposed DKNNR model and the existing MONR model were

analyzed. The estimated transition probabilities ($P_{11}$ and $P_{01}$ in Eq. (3)) as well as the occurrence
probability ($P_1$ in Eq. (5)) are shown in Table 2 and Figure 5 - Figure 7 for the observed data and
the data generated from the DKNNR and MONR models. In Table 2, the observed statistic shows
that $P_{11}$ is always higher than $P_{01}$ and $P_1$ is between $P_{11}$ and $P_{01}$. Site 6 shows the lowest $P_{11}$ and
$P_1$ and site 12 shows the highest $P_{11}$.

As shown in Figure 5, the probability $P_{11}$ of the observed data shows that sites 6, 7, 8, and 9

located in the northern part of the region exhibited lower consistency (i.e. consecutive rainy days)
than did the other sites, while sites 5 and 12 had higher probability of $P_{11}$ than did other sites. Both
models preserved well the observed $P_{11}$ statistic. It seems that the MONR model had a slightly
better performance, since this statistic is parameterized in the model as shown in section 2.2 and
that is the same for P01 and P1 as shown in Figure 6 and Figure 7. Note that the MONR model
employed the transition probabilities in simulating rainfall occurrence, while the DKNNR model
did not. The occurrence probability $P_1$ can be described with the combination of transition
probabilities as in Eq. (5). Even though the transition probabilities were not employed in
simulating rainfall occurrence, the DKNNR model preserved this statistic fairly well.
In the DKNNR modeling procedure, the simple distance measurement in Eq. (11) allows to
preserve transition probabilities in that the following multisite occurrence is resampled from the
historical data whose previous states of multisite occurrence ($x_i^s$) are similar to the current
simulation multisite occurrence ($X_c^s$). This summarized distance ($D_i$) is an essential tool in the
proposed DKNNR modeling. The condition of the current weather system is memorized and the
system is conditioned on simulating the following multisite occurrence with the distance
measurement like a precipitation weather system dynamically changes but often it impacts the
system of the following day.
As shown in Figure 6, the $P_{01}$ probability showed a slightly different behavior such that sites
1, 2, and 3 located in the middle part of the Yeongnam province showed a higher probability than
did other sites. A slight underestimation was seen for sites 2 and 11 but it was not critical, since its
observed value with a cross mark was close to the upper IQR representing 75 percentile.
The behavior of $P_1$ was found to be the same as that of the $P_{11}$ probability. It can be seen in
Figure 7 that no significant underestimation is seen for the DKNNR model (top panel). The $P_1$
statistic was fairly preserved by both DKNNR and MONR models. Note that the MONR model
parameterized the $P_1$ statistic through the transition probabilities as in Eq. (5), while the DKNNR
model did not. Although the DKNNR model used almost no parameters for simulation, the $P_1$
statistic was preserved fairly well.
**6.3. Cross-correlation**
Cross-correlation is a measure of the relationship between sites. The preservation of cross-
correlation is important for the simulation of precipitation occurrence and is required in the
regional analysis for water resources management or agricultural applications. Furthermore,
lagged cross-correlation is also essential as much as is cross-correlation (i.e. contemporaneous
correlation). For example, the amount of streamflow for a watershed from a certain precipitation
event is highly related with lagged cross-correlation.
Daily precipitation occurrence, in general, shows the strongest serial correlation at lag-1 and
its correlation decays as the lag gets longer. This is because a precipitation weather system moves
according to the surrounding pressure and wind direction that dynamically change within a day or
week. Therefore, we analyzed the lag-1 cross-correlation in the current study as the representative
lagged correlation structure.
The cross-correlation of observed data is shown in Table 3. High cross-correlation among
grouped sites, such as sites 6, 7, and 8 (northern part) and sites 3, 4, and 5, as well as 12 (southeast
coastal area, 0.68-0.87), was found. As expected, sites 5 and 12 had the highest cross-correlation
(0.87) due to proximity. The northern sites and coastal sites showed low cross-correlation. This
observed cross-correlation was well preserved in the data generated from both DKNNR and
MONR models, as shown in Figure 8 as well as Table 4 and Table 5. However, consistently slight
but significant underestimation of cross-correlation was seen for the data generated by the MONR
model (see the bottom panel of Figure 8). Note that the errorbars are extended to upper and lower
lines of the circles to 1.95×standard deviation.  The difference of RMSE in Table 6 showed this
characteristic, as most of the values were positive, indicating that the proposed DKNNR model
performed better for cross-correlation.

The lag-1 cross-correlation of observed data, as shown in Table 7, ranged from 0.22-0.35.

The lag-1 cross-correlation for the same site (i.e. $\rho_1(q,s)$, $q=s$) was autocorrelation and was highly
related with $P_{01}$ and $P_{11}$ as in Eq. (6). All the lag-1 cross-correlations exhibited similar magnitudes
even for autocorrelation. This implies that the lag-1 cross-correlation among the selected sites was
as strong as the autocorrelation and as much as the transition probabilities $P_{01}$ and $P_{11}$, thereof.

The observed lag-1 cross-correlations were well preserved in the data generated by the

DKNNR model, as shown in the top panel of Figure 9, while the MONR model showed significant
underestimation, as seen in the bottom panel of Figure 9. The difference of RMSE shown in Table
8 reflects this behavior. In the bottom panel of Figure 9, some of the lag-1 cross-correlations were
well preserved, that were aligned with the base line. From Table 8, the MONR model reproduced
the autocorrelations well with the shaded values. It is because the lag-1 autocorrelation was
indirectly parameterized with the transition probabilities of $P_{11}$ and $P_{01}$ as in Eq. (6). Other than
this autocorrelation, the lag-1 cross-correlation was not reproduced well with the MONR model.
This shortcoming was mentioned by Wilks (1998). Meanwhile, the proposed DKNNR model
preserved this statistic without any parameterization.

We further tested the performance measurements of MAE and Bias whose estimates showed

that MAE had no difference from RMSE. In addition, Bias of lag-1 correlation presented
significant negative values implying its underestimation for the simulated data of the MONR
model as shown in Table 9, while Table 10 of the DKNNR model showed a much smaller bias.
Also, the whole year data instead of the summer season data was tested for model fitting.
Note that all the results presented above were for the summer season data (June-September) as
mentioned in section 4 on data description. The lag-1 cross-correlation is shown in Figure 10 which
indicates that the same characteristic was observed as for the summer season, such that the
proposed DKNNR model preserved better the lagged cross-correlation than did the existing
MONR model. Other statistics, such as correlation matrix and transition probabilities, exhibited
the same results (not shown). Also, other seasons were tried but the estimated correlation matrix
was not a positive semi-definite matrix and its inverse cannot be made for multivariate normal
distribution in the MONR model. It was because the selected stations were close to each other
(around 50-100 km) and produced high cross-correlation, especially in the occurrence during dry
seasons. Special remedy for the existing MONR model should be applied, such as decreasing
cross-correlation by force, but further remedy was not applied in the current study since it was not
within the current scope and focus.
**6.4. Adaptation to climate change**
Model adaptability to climate change in hydro-meteorological simulation models is a critical
factor, since one of the major applications of the models is to assess the impact of climate change.
Therefore, we tested the capability of the proposed model in the current study by adjusting the
probabilities of cross-over and mutation as in Eqs. (15) and (16). A number of variations can be
made with different conditions.
In Figure 11, the changes of transition and marginal probabilities are shown along with the
increase of crossover probability $P_{cr}$ from 0.01 to 0.2 with the condition that that the candidate
value is one and the previous value is also one as in Eq. (15) for the selected 5 stations among the
12 stations (from station 1 to station 5, see Table 1 for details). The stations were limited in this
analysis due to computational time. In each case 100 series were simulated. The average value of
the simulated statistics is presented in the figure. It is obvious that the transition probability $P_{11}$
increased as intended along with the increase of $P_{cr}$. As expected from Eq. (5), $P_1$ presents that the
change of $P_1$ is highly related to $P_{11}$. However, the probability $P_{01}$ fluctuated along with the
increase of $P_{cr}$. Elaborate work to adjust all the probabilities is however required.
The changes in transition and marginal probabilities are presented in Figure 12 with
increasing mutation probability $P_m$ from 0.01 to 0.2 under the condition that the candidate value is
one so that the marginal probability $P_1$ increased. $P_{01}$ also increased along with increasing $P_1$. The
change of P11 was not related with other probabilities. The combination of the adjustment of $P_{cr}$
and $P_m$ with a certain condition to the previous state will allow the specific adaptation for
simulating future climatic scenarios.
As an example, assume that the occurrence probability (P1) of the control period is 0.26 (see
the dotted line with cross on the bottom panel of Figure 11 and Figure 12) and GCM output
indicates that the occurrence probability (P1) increases up to 0.27.  This can be achieved with
increasing either the crossover probability to 0.1 or the mutation probability to 0.05. Note that the
crossover probabilities might affect the stations each other, while the mutation probabilities do not
Climate change, however, may refer to a larger phenomenon, which cannot be addressed
directly through modifying only the marginal and transition probabilities as in the current study.
Further model development on systematically varying temporal and spatial cross-correlations is
required to properly address climate change of the regional precipitation system.

# 7. Conclusions

In the current study, the discrete version of a nonparametric simulation model, based on KNNR, is proposed to overcome the shortcomings of the existing MONR model, such as long stochastic simulation for parameter estimation and underestimation of the lagged crosscorrelation between sites as well as testing the adaptability for climatic change. Occurrence and transition probabilities and cross-correlation as well as lag-1 cross-correlation are estimated for both models. Better preservation of cross-correlation and lag-1 cross-correlation with the DKNNR model than the MONR model is observed. For some cases (i.e., the whole year data and other seasons than the summer season), the estimated cross-correlation matrix is not a positive semi-definite matrix so the multivariate normal simulation is not applicable for the MONR model, because the tested sites are close to each other with high cross-correlation.

Results of this study indicate that the proposed DKNNR model reproduces the occurrence and transition probabilities satisfactorily and preserves the cross-correlations better than does the existing MONR model. Furthermore, not much effort is required to estimate the parameters in the DKNNR model, while the MONR model requires a long stochastic simulation just to estimate each parameter. Thus, the proposed DKNNR model can be a good alternative for simulating multisite precipitation occurrence.

We tested further the enhancement of the proposed model for adapting to climate change by modifying the mutation and crossover probabilities $P_m$ and $P_{cr}$. Results show that the proposed DKNNR model has the capability to adapt to the climate change scenarios, but further elaborate work is required to find the best probability estimation for climate change. Also, only the marginal and transition probabilities cannot address the climate change of regional precipitation. The

variation of temporal and spatial cross-correlation structure must be considered to properly address
the climate change of the regional precipitation system. Further study on improving the model
adaptability to climate change will be followed in the near future. Also, the simulated multisite
occurrence can be coupled with a multisite amount model to produce precipitation events,
including zero values. Further development can be made for multisite amount models with a
nonparametric technique, such as KNNR and bootstrapping.
**Code and Data Availability**
DKNNR code is written in Matlab and is available as a supplement.
The precipitation data employed in the current study is downloadable through
http://www.weather.go.kr/weather/main.jsp
**Author Contribution**
T. Lee and V. Singh conceived of the presented idea. T. Lee developed the theory and
programming. V. Singh supervised the findings of the current work and the writing manuscript.
**Acknowledgment**
This work was supported by the National Research Foundation of Korea (NRF) grant (NRF-
2018R1A2B6001799) funded by the Korean Government (MEST).
# Appendix A: Example of DKNNR
In this appendix, one example of DKNNR simulation is presented with observed dataset in
Table A 1 (i.e. $\mathbf{x}_i = [x_i^s]_{s \in \{1,S\}}$ for $i=1,\ldots,n$; here $S=12$ and $n=16$). The upper part of the table
presents the observed precipitation (unit: mm). Its occurrence data is presented in the bottom part
of this table. The current precipitation occurrence $\mathbf{X}_c = [X_c^s]_{s\in\{1,...,12\}}$ is shown in the second row of
Table A 2. The number of nearest neighbors $k = \sqrt{n} = \sqrt{16} = 4$ and the parameters for GA (i.e. $P_c$
and $P_m$) are 0.1 and 0.01, respectively. Simulation can be made as follows:

506    (1) Estimate the distance $D_i$ between $\mathbf{x}_i$ and $\mathbf{X}_c$ for $i=1,...,n$-1 as in Eq.(11). For example,

507     for $i$=1,

$$D_1 = \sum_{s=1}^{S}\left|X_c^s - x_1^s\right| = |0-1| + |1-1| + ... + |0-1| = 6$$

509     All the estimated distances are shown in the last column of Table A 2.

510    (2) The daily index values are sorted according to the smallest distances shown in the first

511     two columns of Table A 3. The sorted day indices and their corresponding distances are

512     shown in the third and fourth columns of Table A 3. From the $k$ number of sorted indices,

513     one is selected with the weight probability (see Eq.(12)), which is shown in the last

514     column of Table A 3.

515    (3) Simulate a uniform random number ($u$) between 0 and 1. Say $u$=0.321. This value must

516     be compared with the cumulative weighted probabilities in the last column of Table A 3

517     as [0 0.48 0.72 0.88 1.0]. The corresponding day index is assigned as to where the

518     simulated uniform number falls in the cumulative weighted probabilities, here [0 0.48].

519     Therefore, the selected day, $p$, is 14. The occurrences of the following day $p$+1=15 for 12

520     stations are selected as in the second row of Table A 4.

521    (4) For GA mixture, another set must be chosen as in step (3). Say $u$=0.561, which falls in

522     [0.48 0.72]. The second one should be selected. However, there are a number of days with

the same distances. Specifically, six days have the same distances with $D_i$=4. In this case,

one among all six days is selected with equal probability. Assume that $p$=4 is selected and

the following occurrences are selected, as shown in the third row of Table A 4.

(5) With two sets, crossover and mutation process is performed as follows:

(5-1)  Crossover: For each station, a uniform random number ($\varepsilon$) is generated and

compared with $P_c$=0.1 here. Say $\varepsilon$ =0.345, then skip since $\varepsilon$ =0.345> $P_c$=0.1. For

$s$=6, assume the generated random number, $\varepsilon$ (=0.051)< $P_c$(=0.1) and then switch

the $6^{th}$ station value of Set 1 into the value of Set 2 (see Table A 4). The occurrence

state of $X_{c+1}^s$ turns into 1 from 0 as shown in the fourth row of Table A 4 as well as

station 8.

(5-2) Mutation: For each station, a uniform random number ($\varepsilon$) is generated and compared

with $P_m$=0.01. For $s$=12, assume $\varepsilon$ =0.009< $P_m$=0.01 and switch the $12^{th}$ station

value of Set 1 with the one selected among all the observed $12^{th}$ station values with

equal probability (here the last column, $s$=12, of the bottom part of Table A 1, [1 1

0 0 … 1]). The occurrence state of $X_{c+1}^{12}$ turns into 0 from 1 as shown in the fourth

column of Table A 4.

(6) Repeat steps (1)-(5) until the target simulation length is reached.

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

Table 1. Information on 12 selected stations from Yeongnam province, South Korea.

| Order | Station Number[†] | Name | Longitude | Latitude |
|---|---|---|---|---|
| 1 | 138 | Pohang | 129.3797 | 36.0327 |
| 2 | 143 | Daegu | 128.6189 | 35.8850 |
| 3 | 152 | Ulsan | 129.3200 | 35.5600 |
| 4 | 159 | Busan | 129.0319 | 35.1044 |
| 5 | 162 | Tongyeong | 128.4356 | 34.8453 |
| 6 | 277 | Youngdeok | 129.4092 | 36.5331 |
| 7 | 278 | Uisung | 128.6883 | 36.3558 |
| 8 | 279 | Gumi | 128.3206 | 36.1306 |
| 9 | 281 | Youngcheon | 128.9514 | 35.9772 |
| 10 | 285 | Hapcheon | 128.1697 | 35.5650 |
| 11 | 288 | Milyang | 128.7439 | 35.4914 |
| 12 | 294 | Geojae | 128.6044 | 34.8881 |

[†]The station number indicates the identification number operated by Korea Meteorological
Administration (KMA).


Table 2. Occurrence and transition probabilities of observed data and data simulated by DKNNR
and MONR for 12 stations from Yeongnam province, South Korea, during the summer season.
Note that 100 sets with the same record length as the observed data were simulated and the
statistics of 100 sets were averaged.

|  | Obs | | | DKNNR | | | MONR | | |
|---|---|---|---|---|---|---|---|---|---|
|  | P11 | P01 | P1 | P11 | P01 | P1 | P11 | P01 | P1 |
| S1 | 0.56 | 0.27 | 0.38 | 0.56 | 0.27 | 0.38 | 0.56 | 0.26 | 0.37 |
| S2 | 0.56 | 0.27 | 0.38 | 0.58 | 0.26 | 0.38 | 0.57 | 0.25 | 0.37 |
| S3 | 0.57 | 0.26 | 0.38 | 0.58 | 0.26 | 0.38 | 0.56 | 0.26 | 0.37 |
| S4 | 0.58 | 0.25 | 0.37 | 0.58 | 0.25 | 0.37 | 0.57 | 0.24 | 0.36 |
| S5 | 0.58 | 0.25 | 0.37 | 0.59 | 0.24 | 0.37 | 0.58 | 0.24 | 0.36 |
| S6 | 0.52 | 0.25 | 0.34 | 0.50 | 0.24 | 0.33 | 0.52 | 0.24 | 0.33 |
| S7 | 0.55 | 0.26 | 0.36 | 0.56 | 0.25 | 0.36 | 0.55 | 0.24 | 0.35 |
| S8 | 0.56 | 0.25 | 0.37 | 0.57 | 0.25 | 0.37 | 0.57 | 0.24 | 0.36 |
| S9 | 0.55 | 0.25 | 0.36 | 0.55 | 0.24 | 0.35 | 0.55 | 0.24 | 0.35 |
| S10 | 0.58 | 0.25 | 0.38 | 0.59 | 0.24 | 0.37 | 0.57 | 0.23 | 0.35 |
| S11 | 0.57 | 0.25 | 0.36 | 0.58 | 0.24 | 0.36 | 0.56 | 0.24 | 0.35 |
| S12 | 0.59 | 0.25 | 0.38 | 0.59 | 0.25 | 0.38 | 0.59 | 0.25 | 0.37 |


Table 3. Cross-correlation of observed data for 12 stations from Yeongnam province, South
Korea.

|  | S1 | S2 | S3 | S4 | S5 | S6 | S7 | S8 | S9 | S10 | S11 | S12 |
|---|---|---|---|---|---|---|---|---|---|---|---|---|
| S1 | 1.00 | 0.70 | 0.70 | 0.64 | 0.58 | 0.70 | 0.65 | 0.63 | 0.75 | 0.64 | 0.66 | 0.59 |
| S2 | 0.70 | 1.00 | 0.67 | 0.64 | 0.61 | 0.64 | 0.70 | 0.72 | 0.79 | 0.72 | 0.74 | 0.62 |
| S3 | 0.70 | 0.67 | 1.00 | 0.75 | 0.68 | 0.61 | 0.57 | 0.57 | 0.68 | 0.67 | 0.74 | 0.70 |
| S4 | 0.64 | 0.64 | 0.75 | 1.00 | 0.79 | 0.56 | 0.56 | 0.55 | 0.65 | 0.66 | 0.73 | 0.82 |
| S5 | 0.58 | 0.61 | 0.68 | 0.79 | 1.00 | 0.51 | 0.54 | 0.55 | 0.61 | 0.65 | 0.70 | 0.87 |
| S6 | 0.70 | 0.64 | 0.61 | 0.56 | 0.51 | 1.00 | 0.69 | 0.65 | 0.68 | 0.59 | 0.59 | 0.54 |
| S7 | 0.65 | 0.70 | 0.57 | 0.56 | 0.54 | 0.69 | 1.00 | 0.78 | 0.71 | 0.65 | 0.63 | 0.55 |
| S8 | 0.63 | 0.72 | 0.57 | 0.55 | 0.55 | 0.65 | 0.78 | 1.00 | 0.71 | 0.68 | 0.65 | 0.56 |
| S9 | 0.75 | 0.79 | 0.68 | 0.65 | 0.61 | 0.68 | 0.71 | 0.71 | 1.00 | 0.68 | 0.71 | 0.62 |
| S10 | 0.64 | 0.72 | 0.67 | 0.66 | 0.65 | 0.59 | 0.65 | 0.68 | 0.68 | 1.00 | 0.77 | 0.66 |
| S11 | 0.66 | 0.74 | 0.74 | 0.73 | 0.70 | 0.59 | 0.63 | 0.65 | 0.71 | 0.77 | 1.00 | 0.70 |
| S12 | 0.59 | 0.62 | 0.70 | 0.82 | 0.87 | 0.54 | 0.55 | 0.56 | 0.62 | 0.66 | 0.70 | 1.00 |


Table 4. Averaged cross-correlation of the 100 simulated series from the DKNNR model for 12
stations from Yeongnam province, South Korea.

|      | S1   | S2   | S3   | S4   | S5   | S6   | S7   | S8   | S9   | S10  | S11  | S12  |
|------|------|------|------|------|------|------|------|------|------|------|------|------|
| S1   | 1.00 | 0.68 | 0.69 | 0.64 | 0.60 | 0.69 | 0.64 | 0.62 | 0.73 | 0.63 | 0.65 | 0.61 |
| S2   | 0.68 | 1.00 | 0.67 | 0.63 | 0.62 | 0.63 | 0.68 | 0.72 | 0.77 | 0.74 | 0.73 | 0.63 |
| S3   | 0.69 | 0.67 | 1.00 | 0.74 | 0.69 | 0.60 | 0.58 | 0.59 | 0.66 | 0.68 | 0.74 | 0.70 |
| S4   | 0.64 | 0.63 | 0.74 | 1.00 | 0.79 | 0.55 | 0.55 | 0.56 | 0.62 | 0.65 | 0.71 | 0.81 |
| S5   | 0.60 | 0.62 | 0.69 | 0.79 | 1.00 | 0.51 | 0.56 | 0.58 | 0.60 | 0.66 | 0.70 | 0.86 |
| S6   | 0.69 | 0.63 | 0.60 | 0.55 | 0.51 | 1.00 | 0.68 | 0.64 | 0.65 | 0.59 | 0.58 | 0.53 |
| S7   | 0.64 | 0.68 | 0.58 | 0.55 | 0.56 | 0.68 | 1.00 | 0.78 | 0.69 | 0.65 | 0.63 | 0.56 |
| S8   | 0.62 | 0.72 | 0.59 | 0.56 | 0.58 | 0.64 | 0.78 | 1.00 | 0.70 | 0.69 | 0.67 | 0.58 |
| S9   | 0.73 | 0.77 | 0.66 | 0.62 | 0.60 | 0.65 | 0.69 | 0.70 | 1.00 | 0.67 | 0.69 | 0.60 |
| S10  | 0.63 | 0.74 | 0.68 | 0.65 | 0.66 | 0.59 | 0.65 | 0.69 | 0.67 | 1.00 | 0.77 | 0.67 |
| S11  | 0.65 | 0.73 | 0.74 | 0.71 | 0.70 | 0.58 | 0.63 | 0.67 | 0.69 | 0.77 | 1.00 | 0.71 |
| S12  | 0.61 | 0.63 | 0.70 | 0.81 | 0.86 | 0.53 | 0.56 | 0.58 | 0.60 | 0.67 | 0.71 | 1.00 |


Table 5. Averaged cross-correlation of 100 simulated series from the MONR model for 12
stations from Yeongnam province.

|      | S1   | S2   | S3   | S4   | S5   | S6   | S7   | S8   | S9   | S10  | S11  | S12  |
|------|------|------|------|------|------|------|------|------|------|------|------|------|
| S1   | 1.00 | 0.63 | 0.67 | 0.58 | 0.54 | 0.66 | 0.62 | 0.60 | 0.68 | 0.55 | 0.62 | 0.53 |
| S2   | 0.63 | 1.00 | 0.61 | 0.60 | 0.57 | 0.59 | 0.68 | 0.68 | 0.75 | 0.66 | 0.72 | 0.58 |
| S3   | 0.67 | 0.61 | 1.00 | 0.71 | 0.67 | 0.57 | 0.56 | 0.53 | 0.65 | 0.61 | 0.71 | 0.69 |
| S4   | 0.58 | 0.60 | 0.71 | 1.00 | 0.78 | 0.50 | 0.52 | 0.52 | 0.61 | 0.62 | 0.69 | 0.78 |
| S5   | 0.54 | 0.57 | 0.67 | 0.78 | 1.00 | 0.48 | 0.51 | 0.53 | 0.57 | 0.62 | 0.67 | 0.81 |
| S6   | 0.66 | 0.59 | 0.57 | 0.50 | 0.48 | 1.00 | 0.67 | 0.62 | 0.63 | 0.54 | 0.54 | 0.49 |
| S7   | 0.62 | 0.68 | 0.56 | 0.52 | 0.51 | 0.67 | 1.00 | 0.75 | 0.70 | 0.61 | 0.62 | 0.52 |
| S8   | 0.60 | 0.68 | 0.53 | 0.52 | 0.53 | 0.62 | 0.75 | 1.00 | 0.66 | 0.64 | 0.61 | 0.52 |
| S9   | 0.68 | 0.75 | 0.65 | 0.61 | 0.57 | 0.63 | 0.70 | 0.66 | 1.00 | 0.63 | 0.69 | 0.57 |
| S10  | 0.55 | 0.66 | 0.61 | 0.62 | 0.62 | 0.54 | 0.61 | 0.64 | 0.63 | 1.00 | 0.72 | 0.61 |
| S11  | 0.62 | 0.72 | 0.71 | 0.69 | 0.67 | 0.54 | 0.62 | 0.61 | 0.69 | 0.72 | 1.00 | 0.66 |
| S12  | 0.53 | 0.58 | 0.69 | 0.78 | 0.81 | 0.49 | 0.52 | 0.52 | 0.57 | 0.61 | 0.66 | 1.00 |


Table 6. The difference of RMSE of cross-correlation between MONR and DKNNR. Note that
the positive value indicates that the DKNNR model better performs in preserving the cross-
correlation, while a negative value (underlined) shows that the MONR model better performs.

| MONR-DKNNR | S1 | S2 | S3 | S4 | S5 | S6 | S7 | S8 | S9 | S10 | S11 | S12 |
|---|---|---|---|---|---|---|---|---|---|---|---|---|
| S1 | 0.000 | 0.014 | 0.004 | 0.013 | 0.012 | 0.012 | 0.008 | 0.005 | 0.024 | 0.031 | 0.011 | 0.035 |
| S2 | 0.014 | 0.000 | 0.023 | 0.013 | 0.021 | 0.009 | 0.010 | 0.013 | 0.018 | 0.027 | 0.011 | 0.020 |
| S3 | 0.004 | 0.023 | 0.000 | 0.015 | 0.004 | 0.014 | 0.003 | 0.022 | 0.009 | 0.028 | 0.011 | 0.004 |
| S4 | 0.013 | 0.013 | 0.015 | 0.000 | 0.002 | 0.017 | 0.018 | 0.014 | 0.018 | 0.018 | 0.027 | 0.024 |
| S5 | 0.012 | 0.021 | 0.004 | 0.002 | 0.000 | 0.014 | 0.021 | 0.014 | 0.015 | 0.013 | 0.015 | 0.012 |
| S6 | 0.012 | 0.009 | 0.014 | 0.017 | 0.014 | 0.000 | 0.006 | 0.010 | 0.030 | 0.018 | 0.029 | 0.021 |
| S7 | 0.008 | 0.010 | 0.003 | 0.018 | 0.021 | 0.006 | 0.000 | 0.005 | 0.008 | 0.024 | 0.012 | 0.023 |
| S8 | 0.005 | 0.013 | 0.022 | 0.014 | 0.014 | 0.010 | 0.005 | 0.000 | 0.032 | 0.019 | 0.022 | 0.023 |
| S9 | 0.024 | 0.018 | 0.009 | 0.018 | 0.015 | 0.030 | 0.008 | 0.032 | 0.000 | 0.019 | 0.005 | 0.027 |
| S10 | 0.031 | 0.027 | 0.028 | 0.018 | 0.013 | 0.018 | 0.024 | 0.019 | 0.019 | 0.000 | 0.020 | 0.021 |
| S11 | 0.011 | 0.011 | 0.011 | 0.027 | 0.015 | 0.029 | 0.012 | 0.022 | 0.005 | 0.020 | 0.000 | 0.022 |
| S12 | 0.035 | 0.020 | 0.004 | 0.024 | 0.012 | 0.021 | 0.023 | 0.023 | 0.027 | 0.021 | 0.022 | 0.000 |

Note that no negative value can be found implying that the DKNNR model preserves the
crosscorrelation better than the MONR model.




Table 7. Lag-1 cross-correlation of observed data for 12 stations from Yeongnam province,
South Korea.

|  | S1 | S2 | S3 | S4 | S5 | S6 | S7 | S8 | S9 | S10 | S11 | S12 |
|---|---|---|---|---|---|---|---|---|---|---|---|---|
| S1 | 0.29[‡] | 0.26 | 0.30 | 0.27 | 0.24 | 0.29 | 0.26 | 0.24 | 0.27 | 0.26 | 0.28 | 0.26 |
| S2 | 0.28 | 0.30 | 0.29 | 0.28 | 0.26 | 0.28 | 0.28 | 0.27 | 0.31 | 0.30 | 0.32 | 0.27 |
| S3 | 0.28 | 0.26 | 0.31 | 0.30 | 0.27 | 0.27 | 0.25 | 0.24 | 0.27 | 0.27 | 0.30 | 0.27 |
| S4 | 0.28 | 0.27 | 0.32 | 0.34 | 0.31 | 0.27 | 0.26 | 0.26 | 0.28 | 0.28 | 0.31 | 0.32 |
| S5 | 0.29 | 0.28 | 0.32 | 0.35 | 0.34 | 0.27 | 0.27 | 0.26 | 0.29 | 0.29 | 0.33 | 0.35 |
| S6 | 0.25 | 0.22 | 0.26 | 0.23 | 0.22 | 0.27 | 0.24 | 0.22 | 0.25 | 0.23 | 0.24 | 0.23 |
| S7 | 0.25 | 0.26 | 0.27 | 0.25 | 0.25 | 0.28 | 0.29 | 0.27 | 0.27 | 0.27 | 0.28 | 0.26 |
| S8 | 0.29 | 0.30 | 0.29 | 0.27 | 0.26 | 0.30 | 0.31 | 0.30 | 0.31 | 0.30 | 0.31 | 0.27 |
| S9 | 0.29 | 0.29 | 0.30 | 0.29 | 0.27 | 0.29 | 0.27 | 0.27 | 0.30 | 0.30 | 0.31 | 0.28 |
| S10 | 0.28 | 0.31 | 0.32 | 0.31 | 0.29 | 0.29 | 0.30 | 0.30 | 0.31 | 0.33 | 0.34 | 0.29 |
| S11 | 0.27 | 0.29 | 0.31 | 0.30 | 0.27 | 0.27 | 0.27 | 0.27 | 0.29 | 0.30 | 0.32 | 0.29 |
| S12 | 0.30 | 0.29 | 0.32 | 0.35 | 0.33 | 0.28 | 0.27 | 0.26 | 0.29 | 0.30 | 0.33 | 0.35 |

[‡]Shaded values represent lag-1 autocorrelation (i.e. the one lagged correlation for the same site).


Table 8. The difference of RMSE of lag-1 cross-correlation between MONR and DKNNR. Note
that a positive value indicates that the DKNNR model better performs in preserving lag-1 cross-
correlation, while a negative value (underlined) shows that the MONR model better performs.

| MONR-DKNNR | S1 | S2 | S3 | S4 | S5 | S6 | S7 | S8 | S9 | S10 | S11 | S12 |
|---|---|---|---|---|---|---|---|---|---|---|---|---|
| S1 | 0.000 | 0.048 | 0.075 | 0.049 | 0.041 | 0.095 | 0.059 | 0.036 | 0.047 | 0.055 | 0.063 | 0.052 |
| S2 | 0.070 | 0.000 | 0.079 | 0.057 | 0.046 | 0.104 | 0.068 | 0.047 | 0.066 | 0.058 | 0.073 | 0.047 |
| S3 | 0.067 | 0.054 | 0.000 | 0.046 | 0.031 | 0.096 | 0.072 | 0.056 | 0.055 | 0.052 | 0.056 | 0.025 |
| S4 | 0.086 | 0.075 | 0.083 | 0.002 | 0.037 | 0.117 | 0.089 | 0.077 | 0.078 | 0.062 | 0.077 | 0.040 |
| S5 | 0.111 | 0.096 | 0.098 | 0.074 | 0.002 | 0.124 | 0.103 | 0.085 | 0.105 | 0.070 | 0.108 | 0.049 |
| S6 | 0.039 | 0.024 | 0.060 | 0.038 | 0.043 | -0.002 | 0.028 | 0.017 | 0.045 | 0.034 | 0.055 | 0.037 |
| S7 | 0.055 | 0.045 | 0.077 | 0.061 | 0.062 | 0.084 | 0.000 | 0.023 | 0.051 | 0.052 | 0.071 | 0.064 |
| S8 | 0.092 | 0.078 | 0.104 | 0.079 | 0.068 | 0.115 | 0.079 | 0.000 | 0.094 | 0.078 | 0.101 | 0.074 |
| S9 | 0.060 | 0.052 | 0.084 | 0.066 | 0.056 | 0.106 | 0.057 | 0.056 | 0.001 | 0.069 | 0.076 | 0.064 |
| S10 | 0.091 | 0.094 | 0.105 | 0.081 | 0.062 | 0.123 | 0.107 | 0.085 | 0.100 | 0.001 | 0.092 | 0.063 |
| S11 | 0.064 | 0.061 | 0.071 | 0.057 | 0.033 | 0.109 | 0.084 | 0.063 | 0.062 | 0.043 | -0.002 | 0.043 |
| S12 | 0.121 | 0.099 | 0.096 | 0.077 | 0.036 | 0.130 | 0.101 | 0.086 | 0.107 | 0.082 | 0.109 | 0.003 |




Table 9. Bias of lag-1 cross-correlation of the generated data from the DKNNR model. Note that
a positive value indicates the overestimation of lag-1 cross-correlation, while a negative value
shows underestimation. Note that $Bias = 1/N \sum_{m=1}^{N} \Gamma_m^G - \Gamma^h$ and see Eq. (18) for the details of each
term.

| | S1 | S2 | S3 | S4 | S5 | S6 | S7 | S8 | S9 | S10 | S11 | S12 |
|---|---|---|---|---|---|---|---|---|---|---|---|---|
| S1 | 0.000 | 0.009 | 0.001 | 0.003 | 0.006 | -0.002 | 0.010 | 0.011 | 0.006 | 0.010 | 0.010 | 0.006 |
| S2 | 0.005 | 0.009 | 0.010 | 0.006 | 0.008 | 0.006 | 0.011 | 0.011 | 0.004 | 0.009 | 0.009 | 0.010 |
| S3 | 0.002 | 0.010 | 0.001 | -0.002 | 0.003 | 0.002 | 0.007 | 0.008 | 0.006 | 0.009 | 0.006 | 0.007 |
| S4 | 0.006 | 0.009 | 0.004 | 0.001 | 0.007 | 0.003 | 0.008 | 0.008 | 0.009 | 0.010 | 0.010 | 0.005 |
| S5 | 0.004 | 0.005 | 0.000 | -0.001 | -0.001 | 0.007 | 0.005 | 0.006 | 0.002 | 0.008 | 0.000 | -0.001 |
| S6 | -0.002 | 0.006 | 0.000 | 0.002 | -0.001 | -0.002 | 0.004 | 0.003 | 0.002 | 0.005 | 0.004 | 0.001 |
| S7 | 0.004 | 0.008 | 0.003 | 0.003 | 0.001 | 0.004 | 0.002 | 0.006 | 0.007 | 0.007 | 0.007 | 0.002 |
| S8 | 0.000 | 0.005 | 0.004 | 0.001 | 0.004 | -0.003 | -0.003 | 0.000 | 0.001 | 0.004 | 0.006 | 0.003 |
| S9 | 0.005 | 0.007 | 0.006 | 0.003 | 0.006 | 0.004 | 0.010 | 0.007 | 0.004 | 0.007 | 0.006 | 0.007 |
| S10 | 0.003 | 0.005 | 0.001 | -0.001 | -0.001 | 0.001 | 0.001 | 0.001 | 0.003 | 0.000 | 0.002 | 0.001 |
| S11 | 0.010 | 0.010 | 0.008 | 0.004 | 0.008 | 0.009 | 0.009 | 0.009 | 0.010 | 0.010 | 0.011 | 0.008 |
| S12 | 0.003 | 0.006 | 0.001 | -0.001 | 0.004 | 0.003 | 0.008 | 0.008 | 0.005 | 0.005 | 0.002 | 0.001 |



Table 10. Bias of lag-1 cross-correlation of the generated data from the Wilks model. Note that a
positive value indicates the overestimation of lag-1 cross-correlation, while a negative value
shows underestimation.

|  | S1 | S2 | S3 | S4 | S5 | S6 | S7 | S8 | S9 | S10 | S11 | S12 |
|---|---|---|---|---|---|---|---|---|---|---|---|---|
| S1 | -0.001 | -0.062 | -0.089 | -0.063 | -0.055 | -0.106 | -0.074 | -0.052 | -0.060 | -0.070 | -0.080 | -0.067 |
| S2 | -0.084 | 0.000 | -0.096 | -0.072 | -0.061 | -0.117 | -0.083 | -0.063 | -0.079 | -0.072 | -0.089 | -0.063 |
| S3 | -0.080 | -0.070 | 0.001 | -0.059 | -0.043 | -0.110 | -0.086 | -0.072 | -0.069 | -0.066 | -0.071 | -0.037 |
| S4 | -0.100 | -0.090 | -0.097 | -0.001 | -0.048 | -0.129 | -0.103 | -0.093 | -0.093 | -0.077 | -0.092 | -0.051 |
| S5 | -0.125 | -0.110 | -0.111 | -0.087 | -0.001 | -0.138 | -0.117 | -0.100 | -0.118 | -0.084 | -0.121 | -0.060 |
| S6 | -0.053 | -0.037 | -0.074 | -0.051 | -0.057 | -0.001 | -0.039 | -0.030 | -0.060 | -0.047 | -0.070 | -0.049 |
| S7 | -0.068 | -0.058 | -0.091 | -0.077 | -0.077 | -0.098 | -0.002 | -0.038 | -0.065 | -0.065 | -0.086 | -0.079 |
| S8 | -0.106 | -0.091 | -0.119 | -0.094 | -0.084 | -0.128 | -0.093 | 0.001 | -0.108 | -0.091 | -0.116 | -0.088 |
| S9 | -0.074 | -0.064 | -0.098 | -0.080 | -0.070 | -0.119 | -0.072 | -0.070 | -0.001 | -0.082 | -0.091 | -0.078 |
| S10 | -0.105 | -0.107 | -0.120 | -0.096 | -0.075 | -0.136 | -0.119 | -0.097 | -0.113 | -0.001 | -0.106 | -0.076 |
| S11 | -0.078 | -0.074 | -0.085 | -0.070 | -0.047 | -0.123 | -0.097 | -0.077 | -0.076 | -0.056 | -0.001 | -0.057 |
| S12 | -0.134 | -0.112 | -0.108 | -0.088 | -0.046 | -0.142 | -0.116 | -0.101 | -0.121 | -0.095 | -0.122 | 0.000 |



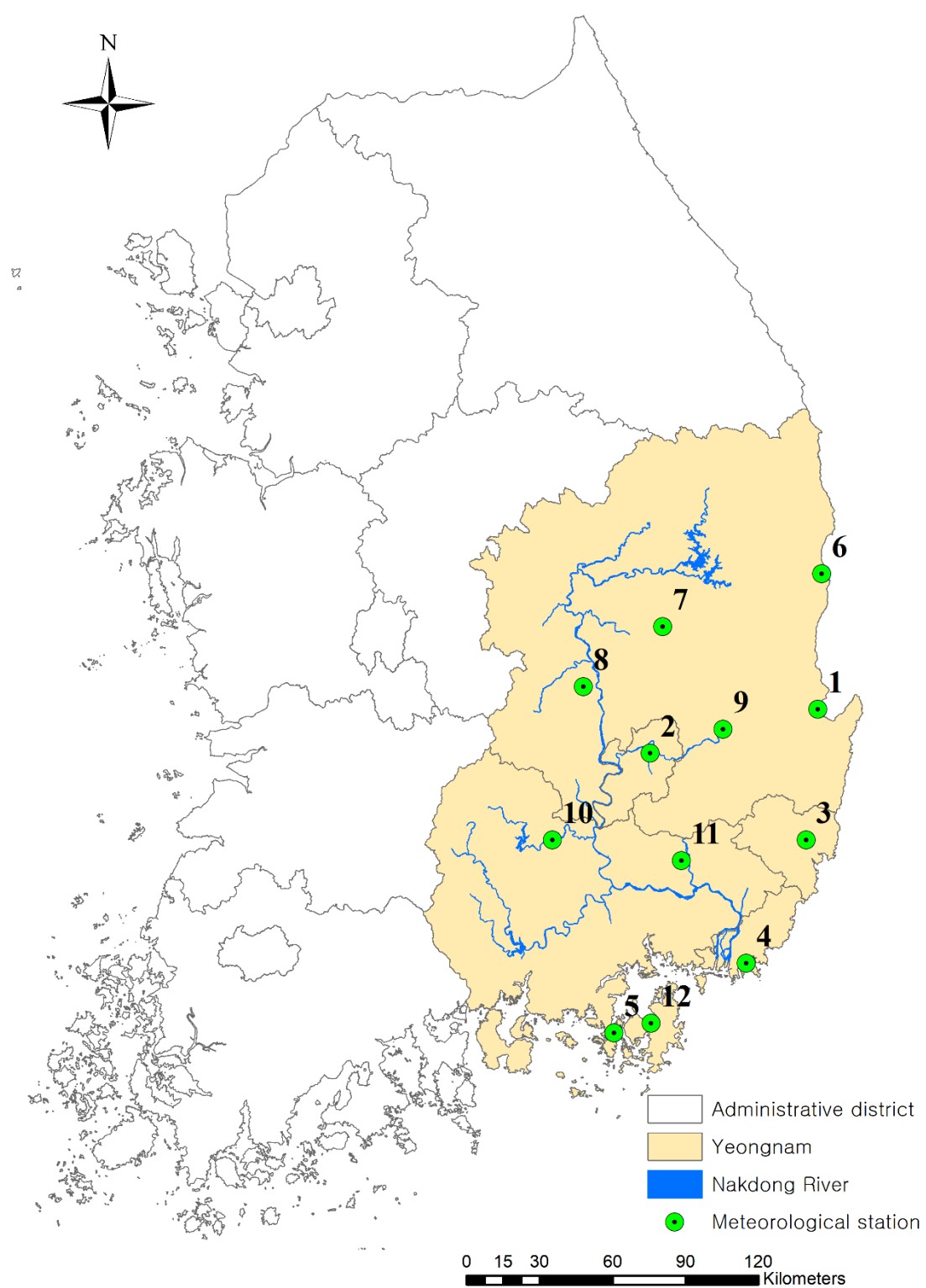


Figure 1. Locations of 12 selected weather stations at the Yeongnam province. See Table 1 for
further information about the stations.

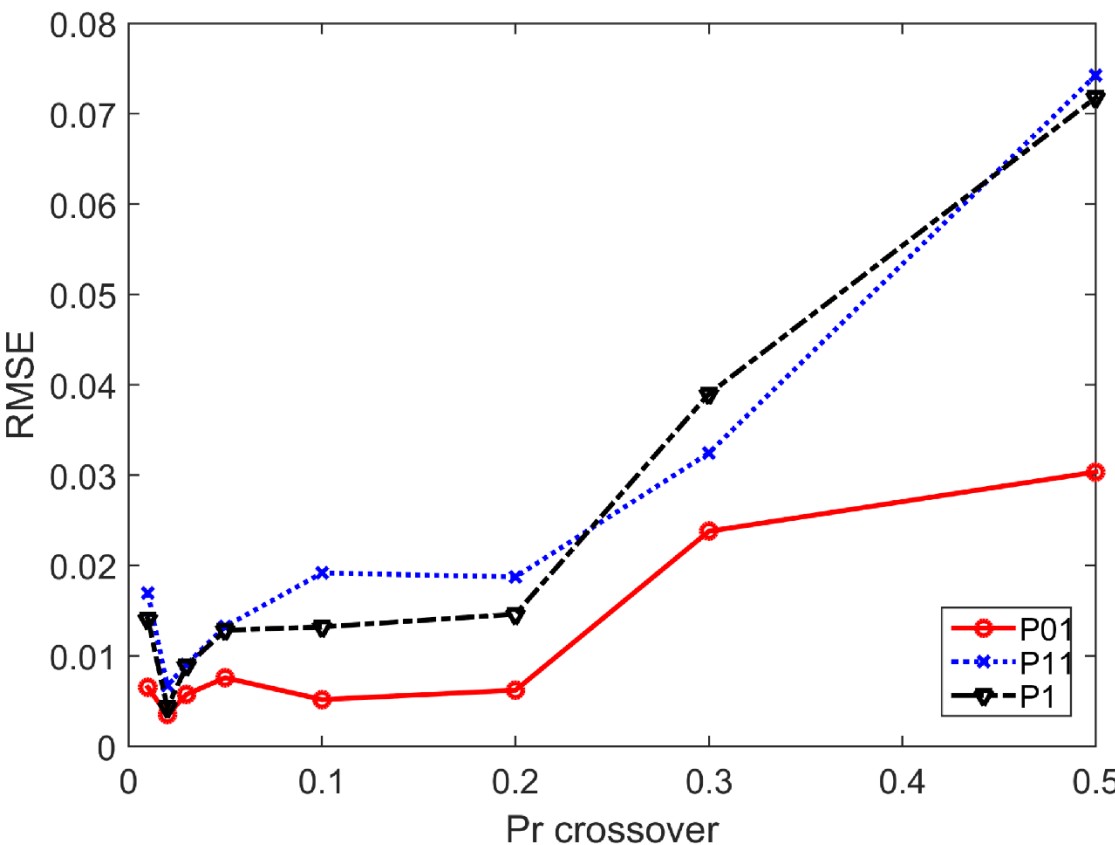

Figure 2. Testing for different probabilities of crossover Pcr. RMSE is estimated for all the tested
12 stations for each transition and limiting probability of the simulated data with the record
length of 100,000.



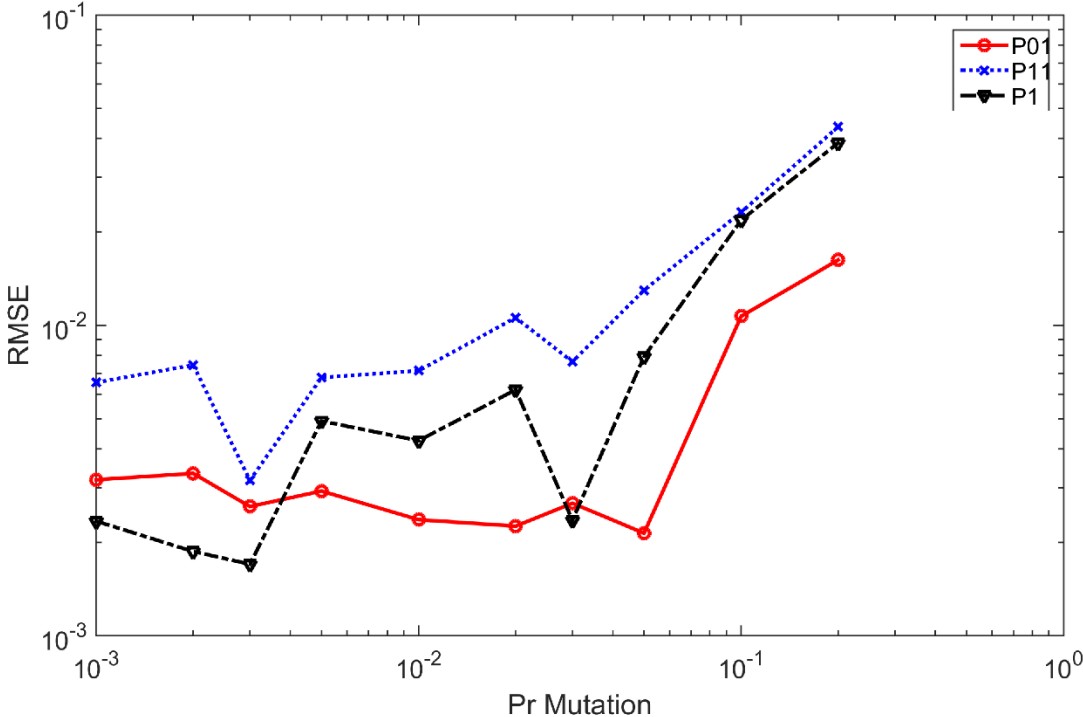


Figure 3. Testing for different probabilities of mutation $P_m$. RMSE is estimated for all the tested
12 stations for each transition and limiting probability of the simulated data with the record
length of 100,000.


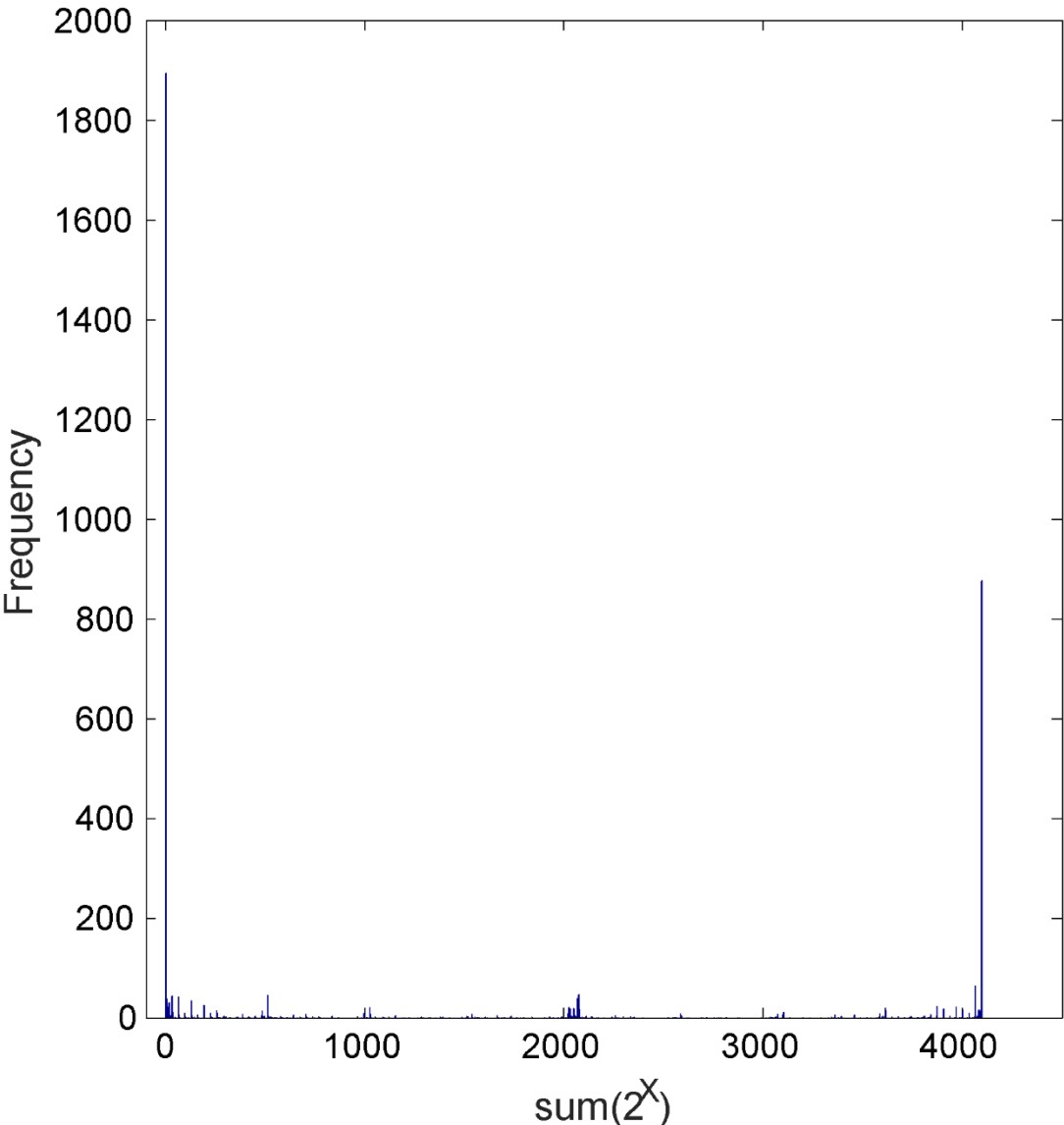

Figure 4. Frequency of the observed patterns among all the possible cases ($2^{12}$=4096). The X
coordinate indicates each pattern with the numbering of the binary number system. All zero (0)
and all one (4095) has the largest and second largest numbers of frequency as 1894 and 877,
respectively as expected meaning all dry and all wet stations. Note that the bars are very sporadic
indicating a number of occurrence patterns are not observed.

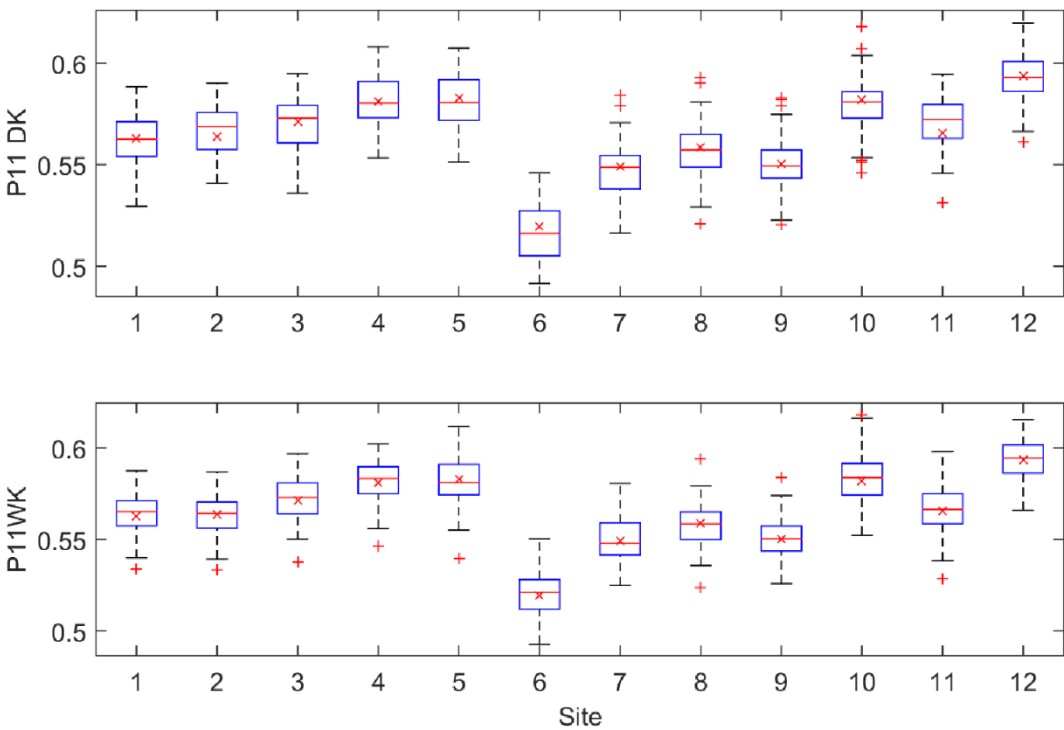


Figure 5. Boxplots of the P11 probability for the simulated data from the DKNNR model (top
panel) and the MONR model (bottom panel) as well as the observed (x marker) for the 12
selected weather stations from the Yeongnam province.

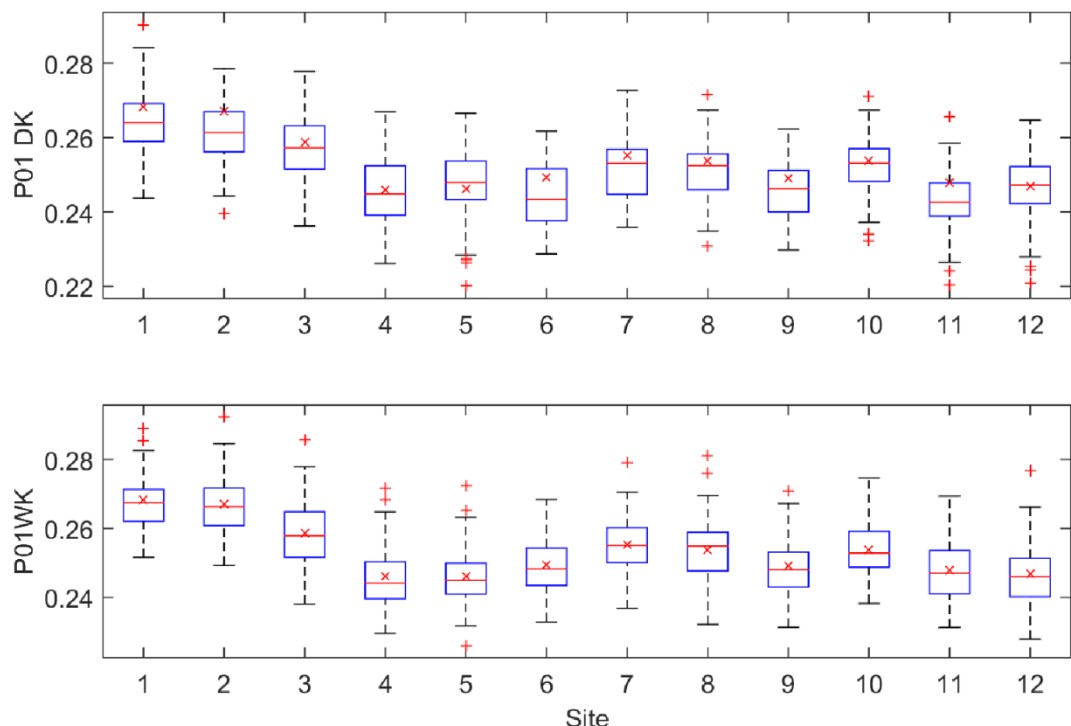


Figure 6. Boxplots of the P01 probability for the data simulated from the DKNNR model (top
panel) and the MONR model (bottom panel) as well as the observed (x marker) for the 12
selected weather stations from the Yeongnam province.










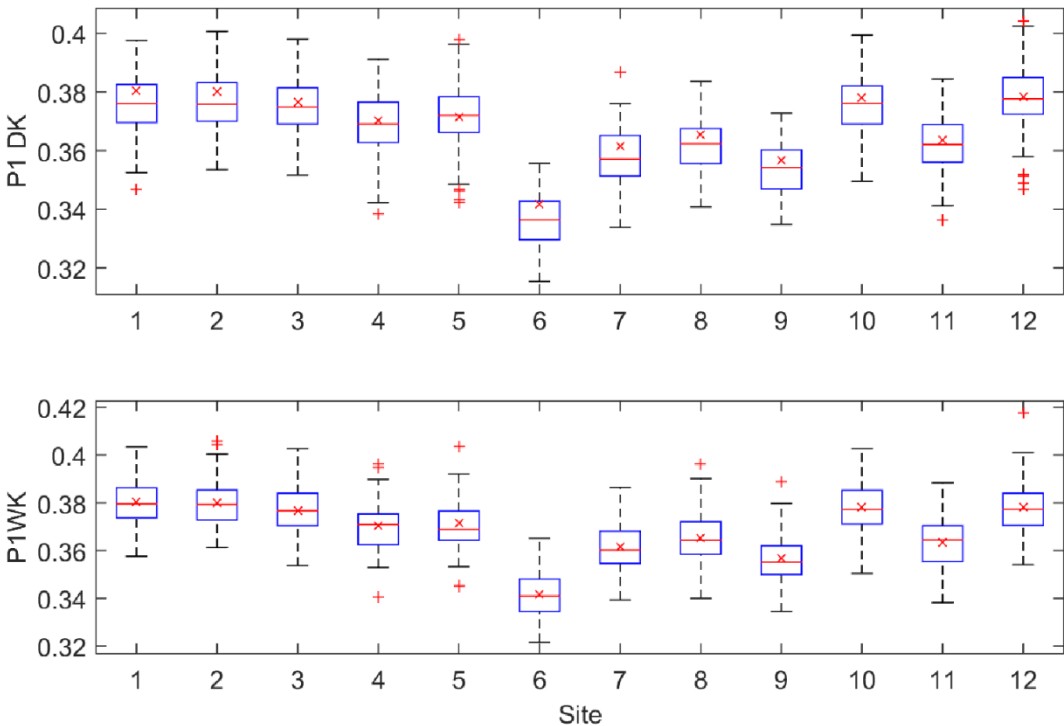


Figure 7. Boxplots of the P1 probability for the data simulated from the DKNNR model (top
panel) and the MONR model (bottom panel) as well as the observed (x marker) for the 12
selected weather stations from the Yeongnam province.

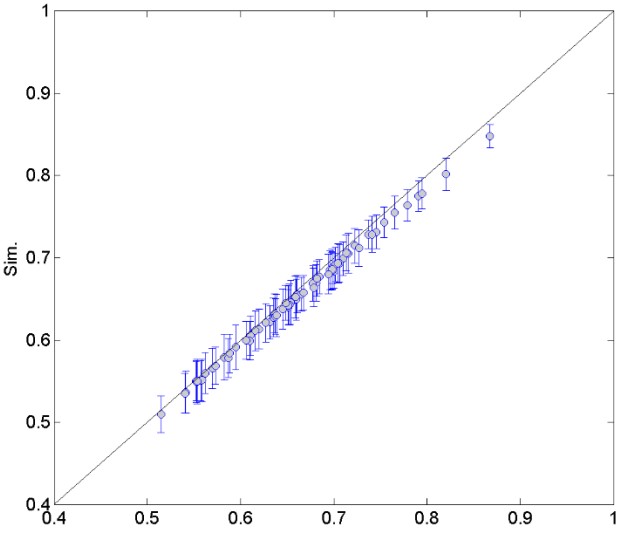

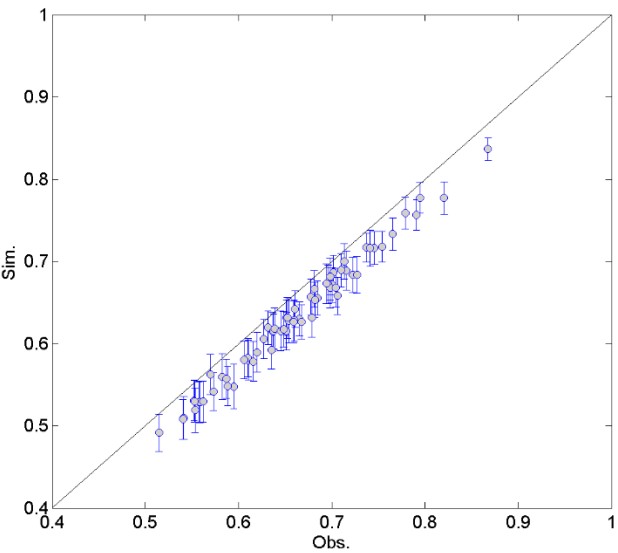


Figure 8. Scatterplot of cross-correlations between 12 weather stations for the observed data (X
coordinate) and the generated data (Y coordinate) generated from the DKNNR model (top panel)
and the MONR model (bottom panel). The cross-correlations from 100 generated series are
averaged for the filled circle and the errorbars upper and lower extended lines indicate the range
of 1.95×standard deviation.


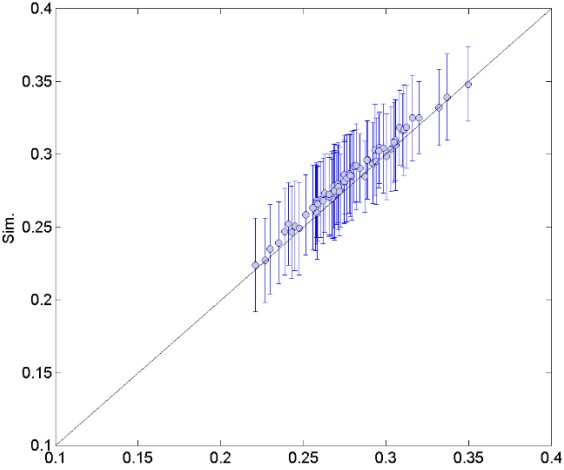

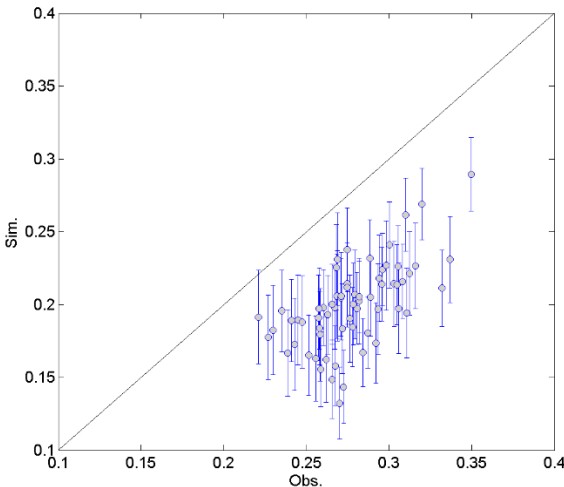


Figure 9. Scatterplot of lag-1 cross-correlations between 12 weather stations for the observed
data (X coordinate) and the generated data (Y coordinate) generated from the DKNNR model
(top panel) and the MONR model (bottom panel). The cross-correlations from 100 generated
series are averaged for the filled circle and the errorbars upper and lower extended lines indicate
the range of 1.95×standard deviation.

764

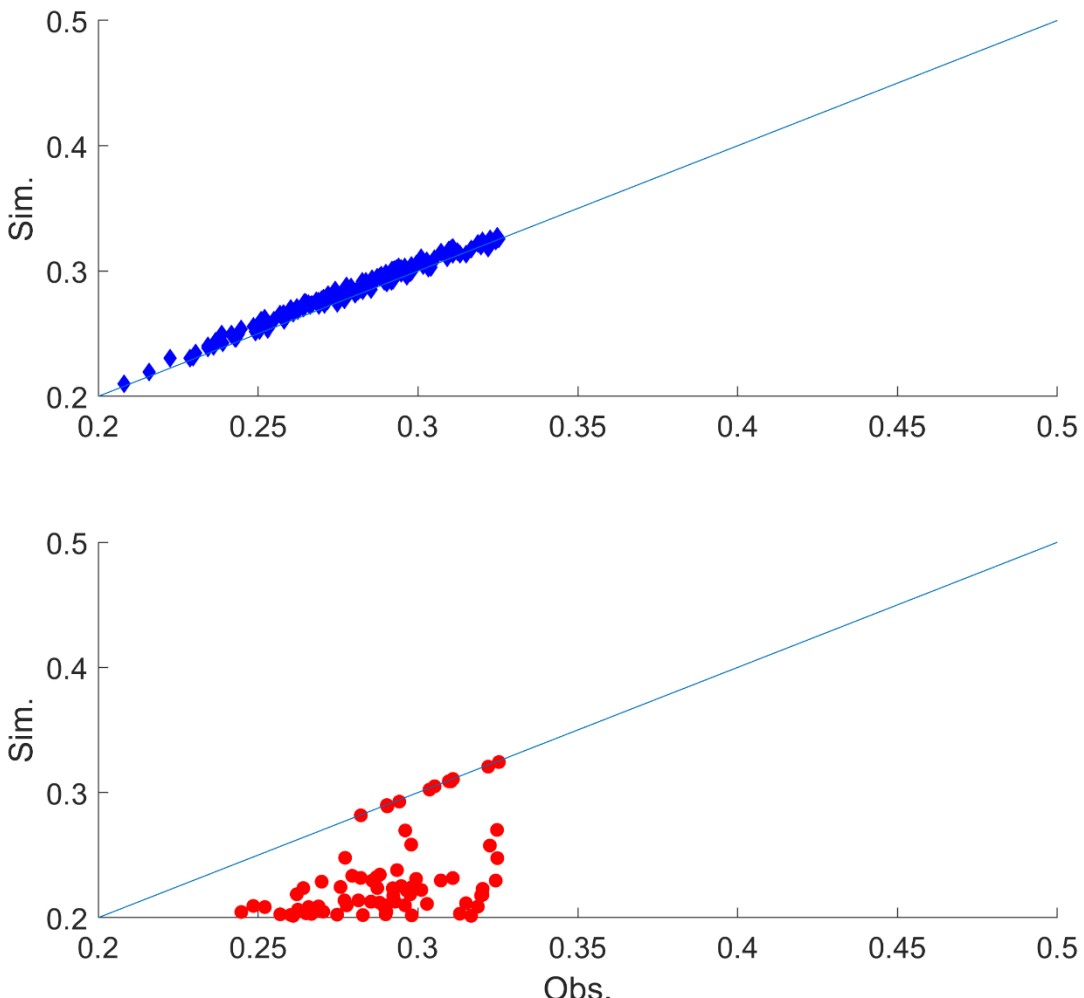

765

Figure 10. Scatterplot of lag-1 cross-correlations between 12 weather stations for the observed data (X coordinate) and the generated data (Y coordinate) generated from the DKNNR model (top panel) and the MONR model (bottom panel) with the whole year data not with the summer season. The cross-correlations from 100 generated series are averaged.






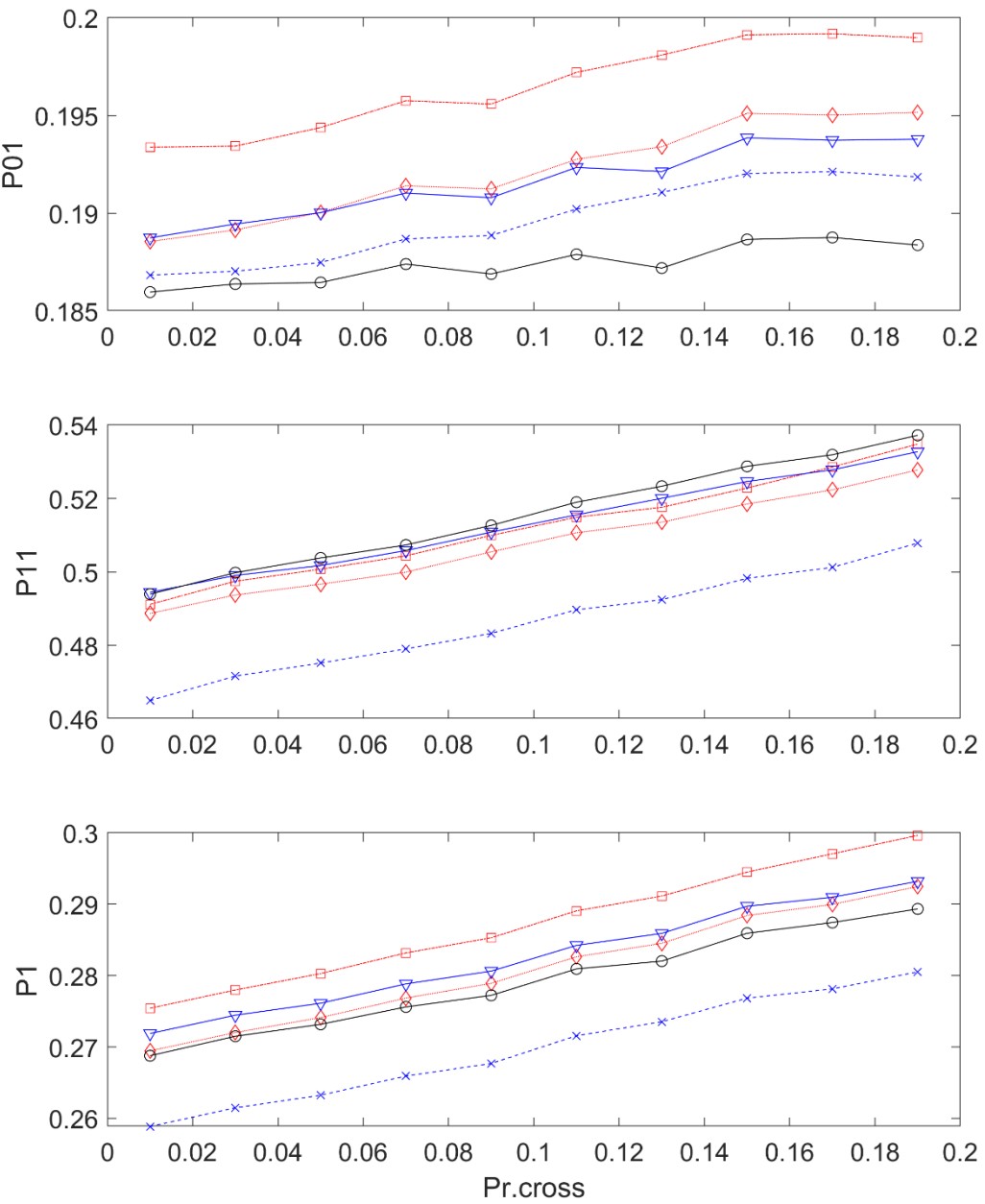


Figure 11. Transition probabilities and marginal distribution for the selected five stations along
with changing the cross-over probability $P_{cr}$ with the condition that the candidate value is one
and the previous value is also one. See Eq.(15) for the detail.


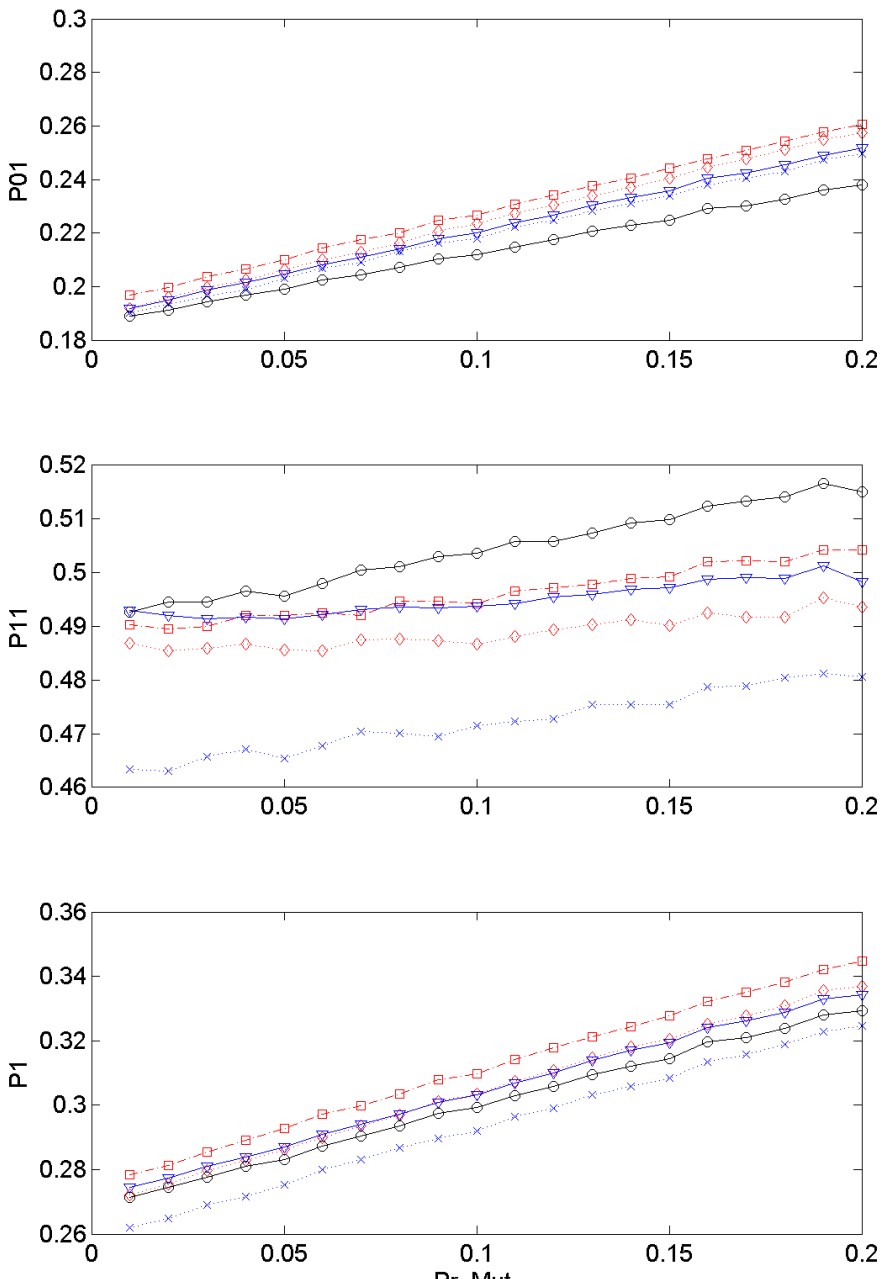


Figure 12. Transition probabilities and marginal distribution along with changing the cross-over
probability with the condition that the mutation is processed only if the candidate value is one.
See Eq.(16) for the detail.



Table A 1. Example dataset of daily rainfall with 12 weather stations and 16 days for measured rainfall (mm) in the upper part of this table and its corresponding occurrences in the bottom part of this table.

| Day | S1 | S2 | S3 | S4 | S5 | S6 | S7 | S8 | S9 | S10 | S11 | S12 |
|-----|------|------|------|------|------|------|------|------|------|------|------|------|
| 1 | 2.0 | 2.9 | 1.2 | 0.0 | 0.0 | 1.8 | 4.0 | 8.9 | 2.0 | 4.6 | 1.3 | 0.6 |
| 2 | 52.6 | 39.8 | 47.2 | 17.4 | 11.8 | 31.0 | 30.0 | 33.7 | 52.0 | 57.8 | 37.0 | 17.5 |
| 3 | 0.0 | 0.0 | 0.0 | 0.0 | 0.0 | 0.0 | 0.0 | 0.0 | 0.0 | 0.0 | 0.0 | 0.0 |
| 4 | 0.0 | 0.0 | 0.0 | 0.0 | 0.0 | 0.0 | 0.0 | 0.0 | 0.0 | 0.0 | 0.0 | 0.0 |
| 5 | 0.2 | 1.0 | 1.4 | 1.9 | 12.3 | 0.0 | 0.0 | 0.0 | 0.7 | 3.1 | 3.5 | 8.1 |
| 6 | 14.8 | 0.2 | 0.8 | 0.2 | 5.0 | 0.0 | 0.0 | 18.0 | 0.0 | 0.0 | 0.6 | 3.1 |
| 7 | 0.0 | 0.0 | 0.0 | 0.0 | 0.0 | 0.0 | 0.0 | 0.0 | 0.0 | 0.0 | 0.0 | 0.0 |
| 8 | 0.0 | 0.0 | 0.0 | 0.0 | 0.0 | 0.0 | 0.0 | 0.0 | 0.0 | 0.0 | 0.0 | 0.0 |
| 9 | 0.0 | 0.0 | 0.0 | 0.0 | 0.0 | 0.0 | 0.0 | 0.0 | 0.0 | 0.0 | 0.0 | 0.0 |
| 10 | 0.0 | 1.0 | 0.0 | 0.4 | 0.0 | 3.8 | 0.0 | 0.1 | 0.0 | 0.0 | 0.0 | 0.0 |
| 11 | 7.1 | 6.4 | 12.8 | 12.8 | 13.6 | 2.3 | 2.0 | 5.4 | 6.0 | 7.3 | 16.4 | 20.3 |
| 12 | 0.0 | 0.0 | 0.0 | 0.0 | 5.5 | 0.0 | 0.0 | 0.0 | 0.0 | 0.0 | 0.0 | 4.3 |
| 13 | 10.0 | 1.6 | 11.6 | 14.3 | 1.5 | 5.4 | 0.0 | 0.0 | 2.5 | 0.0 | 2.7 | 16.1 |
| 14 | 2.3 | 0.0 | 0.7 | 0.0 | 0.0 | 1.4 | 0.0 | 0.0 | 0.0 | 0.0 | 0.0 | 0.0 |
| 15 | 31.5 | 4.3 | 30.6 | 12.7 | 14.4 | 25.8 | 3.5 | 0.8 | 5.0 | 2.7 | 6.5 | 20.3 |
| 16 | 37.0 | 7.8 | 30.1 | 11.2 | 9.6 | 36.8 | 2.5 | 4.7 | 13.5 | 1.7 | 10.1 | 14.1 |
| Day | S1 | S2 | S3 | S4 | S5 | S6 | S7 | S8 | S9 | S10 | S11 | S12 |
| 1 | 1 | 1 | 1 | 0 | 0 | 1 | 1 | 1 | 1 | 1 | 1 | 1 |
| 2 | 1 | 1 | 1 | 1 | 1 | 1 | 1 | 1 | 1 | 1 | 1 | 1 |
| 3 | 0 | 0 | 0 | 0 | 0 | 0 | 0 | 0 | 0 | 0 | 0 | 0 |
| 4 | 0 | 0 | 0 | 0 | 0 | 0 | 0 | 0 | 0 | 0 | 0 | 0 |
| 5 | 1 | 1 | 1 | 1 | 1 | 0 | 0 | 0 | 1 | 1 | 1 | 1 |
| 6 | 1 | 1 | 1 | 1 | 1 | 0 | 0 | 1 | 0 | 0 | 1 | 1 |
| 7 | 0 | 0 | 0 | 0 | 0 | 0 | 0 | 0 | 0 | 0 | 0 | 0 |
| 8 | 0 | 0 | 0 | 0 | 0 | 0 | 0 | 0 | 0 | 0 | 0 | 0 |
| 9 | 0 | 0 | 0 | 0 | 0 | 0 | 0 | 0 | 0 | 0 | 0 | 0 |
| 10 | 0 | 1 | 0 | 1 | 0 | 1 | 0 | 1 | 0 | 0 | 0 | 0 |
| 11 | 1 | 1 | 1 | 1 | 1 | 1 | 1 | 1 | 1 | 1 | 1 | 1 |
| 12 | 0 | 0 | 0 | 0 | 1 | 0 | 0 | 0 | 0 | 0 | 0 | 1 |
| 13 | 1 | 1 | 1 | 1 | 1 | 1 | 0 | 0 | 1 | 0 | 1 | 1 |
| 14 | 1 | 0 | 1 | 0 | 0 | 1 | 0 | 0 | 0 | 0 | 0 | 0 |
| 15 | 1 | 1 | 1 | 1 | 1 | 1 | 1 | 1 | 1 | 1 | 1 | 1 |
| 16 | 1 | 1 | 1 | 1 | 1 | 1 | 1 | 1 | 1 | 1 | 1 | 1 |


Table A 2. Example dataset for estimating distances. The second row presents the current daily
precipitation occurrences for 12 stations and the rows below show the absolute difference
between the current occurrences (**Xc**) and the observed data in Table A 1. The last column
presents the distances in Eq. (11).

| day | S1 | S2 | S3 | S4 | S5 | S6 | S7 | S8 | S9 | S10 | S11 | S12 | Dist |
|-----|----|----|----|----|----|----|----|----|----|-----|-----|-----|------|
| Xc | *0* | *1* | *1* | *0* | *0* | *1* | *1* | *0* | *0* | *0* | *0* | *0* | |
| 1 | 1 | 0 | 0 | 0 | 0 | 0 | 0 | 1 | 1 | 1 | 1 | 1 | **6** |
| 2 | 1 | 0 | 0 | 1 | 1 | 0 | 0 | 1 | 1 | 1 | 1 | 1 | **8** |
| 3 | 0 | 1 | 1 | 0 | 0 | 1 | 1 | 0 | 0 | 0 | 0 | 0 | **4** |
| 4 | 0 | 1 | 1 | 0 | 0 | 1 | 1 | 0 | 0 | 0 | 0 | 0 | **4** |
| 5 | 1 | 0 | 0 | 1 | 1 | 1 | 1 | 0 | 1 | 1 | 1 | 1 | **9** |
| 6 | 1 | 0 | 0 | 1 | 1 | 1 | 1 | 1 | 0 | 0 | 1 | 1 | **8** |
| 7 | 0 | 1 | 1 | 0 | 0 | 1 | 1 | 0 | 0 | 0 | 0 | 0 | **4** |
| 8 | 0 | 1 | 1 | 0 | 0 | 1 | 1 | 0 | 0 | 0 | 0 | 0 | **4** |
| 9 | 0 | 1 | 1 | 0 | 0 | 1 | 1 | 0 | 0 | 0 | 0 | 0 | **4** |
| 10 | 0 | 0 | 1 | 1 | 0 | 0 | 1 | 1 | 0 | 0 | 0 | 0 | **4** |
| 11 | 1 | 0 | 0 | 1 | 1 | 0 | 0 | 1 | 1 | 1 | 1 | 1 | **8** |
| 12 | 0 | 1 | 1 | 0 | 1 | 1 | 1 | 0 | 0 | 0 | 0 | 1 | **6** |
| 13 | 1 | 0 | 0 | 1 | 1 | 0 | 1 | 0 | 1 | 0 | 1 | 1 | **7** |
| 14 | 1 | 1 | 0 | 0 | 0 | 0 | 1 | 0 | 0 | 0 | 0 | 0 | **3** |
| 15 | 1 | 0 | 0 | 1 | 1 | 0 | 0 | 1 | 1 | 1 | 1 | 1 | **8** |
| 16 | 1 | 0 | 0 | 1 | 1 | 0 | 0 | 1 | 1 | 1 | 1 | 1 | **8** |




Table A 3. Example for selecting one sequence for $\mathbf{X_{c+1}}$. The second row presents the distances
in Table A 2. The third and fourth columns show the sorted days and distances for the smallest
distances to the largest in the second column. The fourth row presents the probabilities estimated
with Eq. (12). Note that there are six days whose distances are the same with each other. In this
case all the days are included and among six days, one is selected with equal probabilities.

| Day | Dist. | Sorted Day | Sorted Dist | Prob |
|-----|-------|------------|-------------|------|
| 1 | 6 | 14 | 3 | 0.48 |
| 2 | 8 | 3 | **4** | 0.24 |
| 3 | 4 | 4 | **4** | 0.16 |
| 4 | 4 | 7 | **4** | 0.12 |
| 5 | 9 | 8 | **4** | |
| 6 | 8 | 9 | **4** | |
| 7 | 4 | 10 | **4** | |
| 8 | 4 | 1 | 6 | |
| 9 | 4 | 12 | 6 | |
| 10 | 4 | 13 | 7 | |
| 11 | 8 | 2 | 8 | |
| 12 | 6 | 6 | 8 | |
| 13 | 7 | 11 | 8 | |
| 14 | 3 | 15 | 8 | |
| 15 | 8 | 16 | 8 | |
| 16 | 8 | 5 | 9 | |



Table A 4. Example for GA mixture for $\mathbf{X_{c+1}}$. The second and third rows present two selected
sets, while the third row shows the final set for $\mathbf{X_{c+1}}$ with the crossover at S6 and S8 and the
mutation for S12.

|  | Assigned day, $p$ | Selected day, $p+1$ | S1 | S2 | S3 | S4 | S5 | S6 | S7 | S8 | S9 | S10 | S11 | S12 |
|---|---|---|---|---|---|---|---|---|---|---|---|---|---|---|
| Set1 | 14 | **15** | 1 | 0 | 0 | 1 | 1 | 0 | 0 | 1 | 1 | 1 | 1 | 1 |
| Set2 | 4 | **5** | 1 | 0 | 0 | 1 | 1 | 1 | 1 | 0 | 1 | 1 | 1 | 1 |
| Final |  |  | 1 | 0 | 0 | 1 | 1 | 1 | 0 | 0 | 1 | 1 | 1 | **0** |

