# Peer review of "Discrete k-nearest neighbor resampling for simulating multisite precipitation occurrence and adaption to climate change"

_Geoscientific Model Development, 2018_

## Referee Comment (RC1) · Anonymous Referee #1 · 15 Nov 2018

1. The manuscript presents discrete k-nearest neighbor resampling for simulating multisite precipitation occurrence and adaption to climate change, which is interesting. The subject addressed is within the scope of the journal. 2. However, the manuscript, in its present form, contains several weaknesses. Appropriate revisions to the following points should be undertaken in order to justify recommendation for publication. 3. For readers to quickly catch your contribution, it would be better to highlight major difficulties and challenges, and your original achievements to overcome them, in a clearer way in abstract and introduction. 4. It is shown in the reference list that the authors have several publications in this field. This raises some concerns regarding the potential overlap with their previous works. The authors should explicitly state the novel

contribution of this work, the similarities and the differences of this work with their previous publications. 5. It is mentioned in p.2 that k-nearest neighbor resampling coupling with genetic algorithm is adopted to simulate multisite precipitation occurrence. What are other feasible alternatives? What are the advantages of adopting this particular soft computing technique over others in this case? How will this affect the results? The authors should provide more details on this. 6. It is mentioned in p.2 that multisite occurrence model with standard normal variate is adopted as benchmark for comparison. What are the other feasible alternatives? What are the advantages of adopting this particular model over others in this case? How will this affect the results? More details should be furnished. 7. It is mentioned in p.8 that a random selection procedure is adopted to take into account the cases with the same quantity. What are other feasible alternatives? What are the advantages of adopting this particular procedure over others in this case? How will this affect the results? The authors should provide more details on this. 8. It is mentioned in p.9 that the reproduction procedure in (6-1) is adopted in this study. What are other feasible alternatives? What are the advantages of adopting this particular approach over others in this case? How will this affect the results? The authors should provide more details on this. 9. It is mentioned in p.9 that Eq.(13) is adopted for crossover. What are other feasible alternatives? What are the advantages of adopting this particular crossover type over others in this case? How will this affect the results? The authors should provide more details on this. 10. It is mentioned in p.9 that Eq.(14) is adopted for mutation. What are other feasible alternatives? What are the advantages of adopting this particular mutation type over others in this case? How will this affect the results? The authors should provide more details on this. 11. It is mentioned in p.9 that a simple selection method is adopted for the selection of the number of nearest neighbors. What are other feasible alternatives? What are the advantages of adopting this particular method over others in this case? How will this affect the results? The authors should provide more details on this. 12. It is mentioned in p.11 that 12 weather stations were selected from Yeongnam province are adopted as the case study. What are other feasible alternatives? What are the advantages of adopting this particular case study over others in this case? How will this affect the results? The authors should provide more details on this. 13. It is mentioned in p.11 that historical records of 1976 to 2008 are taken. Why are more recent data not included in the study? Is there any difficulty in obtaining more recent data? Are there any changes to situation in recent years? What are its effects on the result? 14. It is mentioned in p.12 that the root mean square error is adopted to evaluate statistics from 100 generated series. What are the other feasible alternatives? What are the advantages of adopting this particular evaluation metric over others in this case? How will this affect the results? More details should be furnished. 15. It is mentioned in p.16 that "...Special remedy should be applied, such as decreasing cross-correlation by force, but further remedy was not applied in the current study since..." More justification should be furnished on this issue. 16. It is mentioned in p.17 that "...However, the probability $P01$ fluctuated along with the increase of $Pcr$. Elaborate work to adjust all the probabilities is however required..." More justification should be furnished on this issue. 17. Some key parameters are not mentioned. The rationale on the choice of the particular set of parameters should be explained with more details. Have the authors experimented with other sets of values? What are the sensitivities of these parameters on the results? 18. Some assumptions are stated in various sections. Justifications should be provided on these assumptions. Evaluation on how they will affect the results should be made. 19. The discussion section in the present form is relatively weak and should be strengthened with more details and justifications. 20. Moreover, the manuscript could be substantially improved by relying and citing more on recent literatures about contemporary real-life case studies of soft computing techniques in hydrological forecasting such as the followings: ïĄň Fotovatikhah, F., et al., "Survey of Computational Intelligence as Basis to Big Flood Management: Challenges, research directions and Future Work," Engineering Applications of Computational Fluid Mechanics 12 (1): 411-437 2018. ïĄň Wu, C.L., et al., "Rainfall-Runoff Modeling Using Artificial Neural Network Coupled with Singular Spectrum Analysis", Journal of Hydrology 399 (3-4): 394-409 2011. ïĄň Taormina, R., et al., "Neural network river forecasting through

baseflow separation and binary-coded swarm optimization", Journal of Hydrology 529 (3): 1788-1797 2015. ïĄň Wang, W.C., et al., "Improved annual rainfall-runoff forecasting using PSO-SVM model based on EEMD," Journal of Hydroinformatics 15 (4): 1377-1390 2013. ïĄň Cheng, C.T., et al., "Flood control management system for reservoirs," Environmental Modeling & Software 19 (12): 1141-1150 2004. ïĄň Chau, K.W., et al., "Use of Meta-Heuristic Techniques in Rainfall-Runoff Modelling" Water 9(3): article no. 186, 6p 2017. 21. Some inconsistencies and minor errors that needed attention are: ïĄň Replace "...had a slight better..." with "...had a slightly better..." in line 250 of p.13 22. In the conclusion section, the limitations of this study and suggested improvements of this work should be highlighted.

---

## Author Comment (AC1) · 12 Dec 2018

Author response to the reviews of the paper "Discrete k-nearest neighbor resampling for simulating multisite precipitation occurrence and adaption to climate change" (Manuscript # gmd-2018-181-RC1,) Interactive comment on "Discrete k-nearest neighbor resampling for simulating multisite precipitation occurrence and adaption to climate change" by Taesam Lee and Vijay P. Singh

1. The manuscript presents discrete k-nearest neighbor resampling for simulating multisite precipitation occurrence and adaption to climate change, which is interesting. The subject addressed is within the scope of the journal. Reply: The authors appreciate

this reviewer's comment.

2. However, the manuscript, in its present form, contains several weaknesses. Appropriate revisions to the following points should be undertaken in order to justify recommendation for publication. Reply: The authors appreciate the reviewer's comments. the authors improved the quality of the current study according to the given comments. Hope this reviewer is satisfactory to this modification.

3. For readers to quickly catch your contribution, it would be better to highlight major difficulties and challenges, and your original achievements to overcome them, in a clearer way in abstract and introduction. Reply: The authors appreciate the reviewer's comment. Accordingly, the introduction and abstract were improved as follows. Hope the modification is satisfactory.

Abstract: Stochastic weather simulation models are commonly employed in water resources management agricultural applications, forest management, transportation management, and recreational activities. The data simulated by these models, such as precipitation, temperature, and wind, are used as input for hydrological and agricultural models. Stochastic simulation of multisite precipitation occurrence is a challenge because of its intermittent characteristics as well as spatial and temporal crosscorrelation. The multisite occurrence model with standard normal variate (MONR) has been used for preserving key statistics and contemporaneous correlation, but it cannot reproduce lagged crosscorrelation between stations and long stochastic simulation is therefore required to estimate its parameters. Employing a nonparametric technique, k-nearest neighbor resampling (KNNR), and coupling it with Genetic Algorithm (GA), this study proposes a novel simulation method for multisite precipitation occurrence, overcoming the shortcomings of the existing MONR model. The proposed discrete version of KNNR (DKNNR) model is compared with an existing parametric model, called multisite occurrence model with standard normal variate (MONR). The datasets simulated from both the DKNNR model and the MONR model are tested using a number of statistics, such as occurrence and transition probabilities as well as temporal and spa-

tial cross-correlations. Results show that the proposed DKNNR model can be a good alternative for simulating multisite precipitation occurrence, while preserving the lagged crosscorrelation between sites and simulating multisite occurrence from a simple and direct procedure without no parameterization. We also tested the model capability to adapt climate change. It is shown that the model is capable but further improvement is required to have specific variations of the occurrence probability due to climate change. Combining with the generated occurrence, the multisite precipitation amount can then be simulated by any multisite amount model. .

Introduction: Wilks (1998) presented a multisite simulation model for the occurrence process (i.e. X) using the standard normal variable that is spatially dependent, representing the relation between the occurrence variable and the standard normal variable with simulation data. Originally, the occurrence of precipitation had been simulated with discrete Markov Chain (MC) model (Katz, 1977). Compared to the MC model requiring a significant number of parameters to generate multisite occurrence, the multisite occurrence model proposed by Wilks (1998) transforms the standard normal variate and simulates the sequence with multivariate normal distribution, and then back-transforms the multivariate normal sequence to the original domain. The model is able to reproduce the contemporaneous multisite dependence structure and lagged dependence only for the same site while requiring a complex simulation process to estimate parameter for each site and being unable to preserve lagged dependence between sites. Meanwhile, Lee et al. (2010a) proposed a nonparametric-based stochastic simulation model for hydrometeorological variables. They overcame the shortcoming of a previous nonparametric simulation model (Lall and Sharma, 1996), called k-nearest neighbor resampling (KNNR) such that the simulated data cannot produce patterns different from those of the observed data (Brandsma and Buishand, 1998; Mehrotra et al., 2006; St-Hilaire et al., 2012). In addition to this KNNR, Lee et al. (2010a) used a meta-heuristic algorithm Genetic Algorithm (GA) that led to the reproduction of similar populations by mixing the simulated dataset. While the KNNR is employed to find similar historical analogues of multisite occurrence to the current status of a simulation

series, GA is applied to use its skill to generate a new descendant from the historical parent chosen with the KNNR. In this procedure, the multisite occurrence of the precipitation variable can be simulated while preserving spatial and temporal correlations. Note that meta-heuristic techniques to GA have been popularly employed in a number of hydrometeorological applications (Chau, 2017; Fotovatikhah et al., 2018; Taormina et al., 2015; Wang et al., 2013). A number of variants of KNNR-GA have since been applied (Lee et al., 2012; Lee and Park, 2017). None of these models can adopt the multisite occurrence in precipitation whose characteristics are binary and temporally and spatially related. Therefore, in the current study we propose a novel stochastic simulation method for multisite occurrence of the precipitation variable with the KNNR-GA based nonparametric approach that (1) simulates multisite occurrence with a simple and direct procedure without parameterization of all the required occurrence probabilities; and (2) reproduces the complex temporal and spatial correlation between stations as well as the basic occurrence probabilities. Note that the proposed nonparametric model is compared with the most popularly employed model proposed by Wilks (1998). Even though the multisite occurrence data from this model (Wilks, 1998) preserves various statistical characteristics of the observed data well, significant underestimation of lagged cross-correlation still exists. Furthermore, the relation between standard normal variable and occurrence variable relies on long stochastic simulation. The paper is organized as follows. The next section presents a mathematical background of existing multisite occurrence modeling. The modeling procedure is discussed in section 3. The study area and data are reported in section 4. The model is applied in section 5. Results of the proposed model are discussed in section 6, and summary and conclusions are presented in section 7."

4. It is shown in the reference list that the authors have several publications in this field. This raises some concerns regarding the potential overlap with their previous works. The authors should explicitly state the novel contribution of this work, the similarities and the differences of this work with their previous publications. Reply: The authors appreciate th reviewer's thoughtful comment. We have explicitly described the detailed difference and stated the novel contribution of this study as follows

"Meanwhile, Lee et al. (2010a) proposed a nonparametric-based stochastic simulation model for hydrometeorological variables. They overcame the shortcoming of a previous nonparametric simulation model (Lall and Sharma, 1996), called k-nearest neighbor resampling (KNNR) such that the simulated data cannot produce patterns different from those of the observed data (Brandsma and Buishand, 1998; Mehrotra et al., 2006; St-Hilaire et al., 2012). In addition to this KNNR, Lee et al. (2010a) used a meta-heuristic algorithm Genetic Algorithm (GA) that led to the reproduction of similar populations by mixing the simulated dataset. While the KNNR is employed to find similar historical analogues of multisite occurrence to the current status of a simulation series, GA is applied to use its skill to generate a new descendant from the historical parent chosen with the KNNR. In this procedure, the multisite occurrence of the precipitation variable can be simulated while preserving spatial and temporal correlations. Note that meta-heuristic techniques to GA have been popularly employed in a number of hydrometeorological applications (Chau, 2017; Fotovatikhah et al., 2018; Taormina et al., 2015; Wang et al., 2013). A number of variants of KNNR-GA have since been applied (Lee et al., 2012; Lee and Park, 2017). None of these models can adopt the multisite occurrence in precipitation whose characteristics are binary and temporally and spatially related."

5. It is mentioned in p.2 that k-nearest neighbor resampling coupling with genetic algorithm is adopted to simulate multisite precipitation occurrence. What are other feasible alternatives? What are the advantages of adopting this particular soft computing technique over others in this case? How will this affect the results? The authors should provide more details on this. Reply: The authors appreciate the reviewer's comment. While the KNNR is employed to find similar historical analogues of multisite occurrence to the current status of a simulation series, GA is applied to use its skill to generate a new descendant from the historical parent chosen with the KNNR. In this procedure, the multisite occurrence of the precipitation variable can be simulated with preserving spatial and temporal correlations. We added the following in the manuscript accordingly. Note that the location of the page has been changed especially in the introduction section for replying the comment 3 of this reviewer.

"While the KNNR is employed to find similar historical analogues of multisite occurrence to the current status of a simulation series, GA is applied to use its skill to generate a new descendant from the historical parent chosen with the KNNR. In this procedure, the multisite occurrence of the precipitation variable can be simulated while preserving spatial and temporal correlations. Note that meta-heuristic techniques to GA have been popularly employed in a number of hydrometeorological applications (Chau, 2017; Fotovatikhah et al., 2018; Taormina et al., 2015; Wang et al., 2013). A number of variants of KNNR-GA have since been applied (Lee et al., 2012; Lee and Park, 2017). None of these models can adopt the multisite occurrence in precipitation whose characteristics are binary and temporally and spatially related."

6. It is mentioned in p.2 that multisite occurrence model with standard normal variate is adopted as benchmark for comparison. What are the other feasible alternatives? What are the advantages of adopting this particular model over others in this case? How will this affect the results? More details should be furnished. Reply: The authors thanks to this reviewer's comment. Another alternative is to use a multisite version of Markov Chain (M-MC) model by estimating the transition matrix of multisite occurrence. However, this M-MC model requires a number of parameters even difficult to handle and very often no data exist to estimate some of parameters. If this model is applied for comparison, the proposed model shows much better performance than the MONR model. The following is added in the manuscript accordingly.

"Wilks (1998) presented a multisite simulation model for the occurrence process (i.e. X) using the standard normal variable that is spatially dependent, representing the relation between the occurrence variable and the standard normal variable with simulation data. Originally, the occurrence of precipitation had been simulated with discrete Markov Chain (MC) model (Katz, 1977). Compared to the MC model requiring a significant number of parameters to generate multisite occurrence, the multisite occurrence model proposed by Wilks (1998) transforms the standard normal variate and simulates the sequence with multivariate normal distribution, and then back-transforms the multivariate normal sequence to the original domain. The model is able to reproduce the contemporaneous multisite dependence structure and lagged dependence only for the same site while requiring a complex simulation process to estimate parameter for each site and being unable to preserve lagged dependence between sites"

7. It is mentioned in p.8 that a random selection procedure is adopted to take into account the cases with the same quantity. What are other feasible alternatives? What are the advantages of adopting this particular procedure over others in this case? How will this affect the results? The authors should provide more details on this. Reply: The authors appreciate this reviewer's comment. Other than the random selection, one can use always the first one. In such a case, only one historical combination of occurrence will be selected among the combinations with the same distance. The following is added in the manuscript. Hope this modification is satisfactory to this reviewer.

"For example, if S=2 and Xc1=0 and Xc2=1, the two sequences has the same D=1 as [xi1=0 and xi2=0] and [xi1=1 and xi2=1]. In this case, a random selection procedure is required to take into account the cases with the same quantity. One particular time index is randomly selected with the equal probabilities among the time indices of the same distances. Note that instead of the random selection, one can choose always the first one. In such a case, only one historical combination of multisite occurrences will be selected."

8. It is mentioned in p.9 that the reproduction procedure in (6-1) is adopted in this study. What are other feasible alternatives? What are the advantages of adopting this particular approach over others in this case? How will this affect the results? The authors should provide more details on this. Reply: The authors appreciate this reviewer's comment. This reproduction process is a mating process by finding another individual that has similar characteristics with the current one xp+1. With this procedure, a similar vector to the current vector will be mated and produce a new descendant. Alternatively, this procedure can be skipped. Then all the elements of the generated vector will be the same as the historical. The following is added accordingly in the manuscript.

"This reproduction process is a mating process by finding another individual that has similar characteristics to the current one xp+1. With this procedure, a similar vector to the current vector will be mated and produce a new descendant."

9. It is mentioned in p.9 that Eq.(13) is adopted for crossover. What are other feasible alternatives? What are the advantages of adopting this particular crossover type over others in this case? How will this affect the results? The authors should provide more details on this. Reply: The same answer as in the comment 8 can be made to this comment for the feasible alternative. The advantage of this crossover is that a new occurrence vector whose elements are similar to the historical is generated. The following is added in the manuscript accordingly.

"From this crossover, a new occurrence vector whose elements are similar to the historical is generated."

10. It is mentioned in p.9 that Eq.(14) is adopted for mutation. What are other feasible alternatives? what are the advantages of adopting this particular mutation type over others in this case? How will this affect the results? The authors should provide more details on this. Reply: Another alternative can be skipped this procedure. Then always similar multisite occurrence to historical combinations would be generated, which is not feasible for a simulation purpose. The advantage of this mutation is to allow a totally new combination of multisite occurrence to be simulated with this mutation process compared to historical records. The following is added in the manuscript accordingly.

"This mutation procedure allows to generate a multisite occurrence combination that is totally different from the historical records. Without this procedure, always similar multisite occurrences to historical combinations are generated, which is not feasible for a simulation purpose."

11. It is mentioned in p.9 that a simple selection method is adopted for the selection of the number of nearest neighbors. What are other feasible alternatives? What are the advantages of adopting this particular method over others in this case? How will this affect the results? The authors should provide more details on this. Reply: The authors appreciate this reviewer's critical comment. Another alternative is to use generalized cross-validation (GCV) as shown in Sharma and Lall1996 and Lee and Ouarda 2011 by treating this simulation as a prediction problem. However, the current multisite occurrence simulation does not necessarily require accurate value prediction and not much difference on simulation using the simple heuristic approach is reported. Also, this heuristic approach of k selection has been popularly employed for hydrometeorological stochastic simulations (Lall and Sharma, 1996; Lee and Ouarda, 2012; Lee et al., 2010b; Prairie et al., 2006; Rajagopalan and Lall, 1999). The following is added in the manuscript accordingly.

"One can use generalized cross-validation (GCV) as shown in Sharma and Lall1996 and Lee and Ouarda 2011 by treating this simulation as a prediction problem. However, the current multisite occurrence simulation does not necessarily require accurate value prediction and not much difference on simulation using the simple heuristic approach is reported. Also, this heuristic approach of k selection has been popularly employed for hydrometeorological stochastic simulations (Lall and Sharma, 1996; Lee and Ouarda, 2012; Lee et al., 2010b; Prairie et al., 2006; Rajagopalan and Lall, 1999)."

12. It is mentioned in p.11 that 12 weather stations were selected from Yeongnam province are adopted as the case study. What are other feasible alternatives? What are the ad- vantages of adopting this particular case study over others in this case? How will this affect the results? The authors should provide more details on this. Reply: The authors appreciate this reviewer's comment. The object of the current study is to build a simulation model for multisite precipitation occurrence. To validate the proposed model appropriately, tested sites must be highly correlated with each other as well as significant temporal relation. The employed stations inside the Gyeongnam area cover

one of the most important watersheds, the Nakdong River basin, where the Nakdong river pass through the entire basin and its hydrological assessments for agriculture and climate change has particular values in water resources management such as floods and droughts. The following has been added accordingly.

"To validate the proposed model appropriately, tested sites must be highly correlated with each other as well as significant temporal relation. The employed stations inside the Yeongnam area cover one of the most important watersheds, the Nakdong River basin, where the Nakdong river pass through the entire basin and its hydrological assessments for agriculture and climate change has particular values in water resources management such as floods and droughts."

13. It is mentioned in p.11 that historical records of 1976 to 2008 are taken. Why are more recent data not included in the study? Is there any difficulty in obtaining more recent data? Are there any changes to situation in recent years? What are its effects on the result? Reply: The authors appreciate this reviewer's comment. This dataset was employed to illustrate the performance of the proposed model especially for the base period. This dataset has been well evaluated from a number of the previous studies (Lee, 2017). According to this comment, more recent data up to the year2015 whose quality has been checked was added and all the results were modified accordingly. Not much significant difference was found from the results of the previous dataset.

14. It is mentioned in p.12 that the root mean square error is adopted to evaluate statistics from 100 generated series. What are the other feasible alternatives? What are the advantages of adopting this particular evaluation metric over others in this case? How will this affect the results? More details should be furnished. Reply: Another alternative would be MAE and Bias. The estimates showed that MAE has no difference from RMSE and Bias of the lag-1 correlation presents significant negative values implying the underestimation of the lag-1 correlation. The following is added in the manuscript including Table 9 and Table 10.

"We further tested the performance measurements of MAE and Bias. The estimates showed that MAE has no difference from RMSE. In addition, Bias of the lag-1 correlation presents significant negative values implying its underestimation for the simulated data of the MONR model shown in Table 9 while Table 10 of the DKNNR model shows much smaller bias."

15. It is mentioned in p.16 that ": : :Special remedy should be applied, such as decreasing cross-correlation by force, but further remedy was not applied in the current study since: : :" More justification should be furnished on this issue. Reply: The authors appreciate this reviewer's comment. We tried to discuss about the possible improvement of the existing MONR model not the proposed model in the current study. The improvement of the existing model is not within the scope of the current study. Following study can be doable for this issue. The authors consider that no further justification was necessary in the current study since the MONR model has not been proposed in the current study. Hope this reviewer understand this. We improved the sentence as the following to avoid the confusion of the model we discuss.

"Special remedy for the existing MONR model should be applied, such as decreasing cross-correlation by force, but further remedy was not applied in the current study since it was not within the current scope and focus."

16. It is mentioned in p.17 that ": : :However, the probability P01 fluctuated along with the increase of Pcr. Elaborate work to adjust all the probabilities is however required: : :" More justification should be furnished on this issue. Reply: The authors appreciate this reviewer's insightful comment. We agree that more justification and application might be needed to show the capability of the proposed model. However, the current study is focused on proposing a novel approach that simulates multisite occurrence process. Further development for adopting climate change and its application will be presented as a separate work as explained in the conclusion in the following. Hope this reviewer understand the intention of the authors.

Interactive
comment

"We tested further enhancement of the proposed model for adapting climate change through modifying the mutation and crossover probability Pm and Pcr with the current and previous states. The results show that the current model has the capability to adapt to the climate change scenarios, but elaborate work is required however. Further study on improving the model adaptability to climate change will be followed in near future."

17. Some key parameters are not mentioned. The rationale on the choice of the particular set of parameters should be explained with more details. Have the authors experimented with other sets of values? What are the sensitivities of these parameters on the results? Reply: The authors appreciate this reviewer's critical comment. The authors totally agree with this comment. Accordingly, we tested the key parameters for the proposed DKNNR method found that the parameter set of Pcr and Pm as 0.02 and 0.003 shows the best from the result of RMSE estimated with the transition and limiting probabilities of the tested stations. Hope this result is satisfactory to this reviewer. The following is added in the manuscript:

"The roles of crossover probability (Eq. 13) and mutation probability (Eq.14) were studied by Lee et al. [2010a]. In the current study, we further tested to select appropriate parameter set of these two parameters with the simulated data from the DKNNR model and the record length of 100,000. RMSE (Eq. 18) of the transition and limiting probabilities (P11, P01, and P1) between the simulated data and the observed was used since those probabilities are key statistics that the simulated data must be met with the observed and no parameterization on these probabilities has been made for the current DKNNR model. The results are shown in Figure 2 and Figure 3 for Pcr and Pm, respectively. For Pcr in Figure 2, the probability of 0.02 shows the smallest RMSE in all transition and limiting probabilities. The RMSE of Pm in Figure 3 shows slight fluctuation along with Pm. However, all three probabilities have relatively small RMSEs in Pm =0.003. Therefore, the parameter set 0.02 and 0.003 is chosen for Pcr and Pm, respectively and employed in the current study."

Figure 2. Testing for different probabilities of crossover Pcr. RMSE is estimated for all the tested 12 stations for each transition probability.

 

Figure 3. Testing for different probabilities of mutation Pm. RMSE is estimated for all the tested 12 stations for each transition probability.

18. Some assumptions are stated in various sections. Justifications should be provided on these assumptions. Evaluation on how they will affect the results should be made. Reply: The authors appreciate this reviewer's comment. Following the comments from the above, we tried our best to show how the assumption may affect the results. Hope the modification following the previous comment meet this reviewer's expectation.

19. The discussion section in the present form is relatively weak and should be strengthened with more details and justifications. Reply: The authors appreciate this reviewer's critical comment. The discussion has been intensified at the conclusion section. Hope this modification is satisfactory to this reviewer. Note that there is no separate discussion section in the current manuscript. If this reviewer implies other specific section, please let us know.

20. Moreover, the manuscript could be substantially improved by relying and citing more on recent literatures about contemporary real-life case studies of soft computing techniques in hydrological forecasting such as the followings: ïAËŻnËĞ Fotovatikhah, F., et al., "Survey of Computational Intelligence as Basis to Big Flood Management: Challenges, research directions and Future Work," Engineering Applications of Computational Fluid Mechanics 12 (1): 411-437 2018. (Fotovatikhah et al., 2018) ïAËŻnËĞ Wu, C.L., et al., "Rainfall-Runoff Modeling Using Artificial Neural Network Coupled with Singular Spectrum Analysis", Journal of Hydrology 399 (3-4): 394-409 2011. (Wu and Chau, 2011) ïAËŻnËĞ Taormina, R., et al., "Neural network river forecasting through optimization", Journal of Hydrology 529 (3): 1788-1797 2015. (Taormina et al., 2015) ïAËŻnËĞ Wang, W.C., et al., "Improved annual rainfall-runoff forecasting using PSO-SVM model based on EEMD," Journal of Hydroinformatics 15 (4): 1377-1390 2013.

(Wang et al., 2013) ïAËŻnËĞ Cheng, C.T., et al., "Flood control management system for reservoirs," Environmental Modeling & Software 19 (12): 1141-1150 2004.(Cheng and Chau, 2004) ïAËŻnËĞ Chau, K.W.,et al., "Use of Meta-Heuristic Techniques in Rainfall-Runoff Modelling" Water 9(3): article no. 186, 6p 2017. 21. (Chau, 2017) Reply: The authors appreciate relevant works. Almost all the suggested papers that are relevant with this study were included in the current study as the following:

"In this procedure, the multisite occurrence of the precipitation variable can be simulated with preserving spatial and temporal correlations. Note that meta-heuristic techniques to GA have been popularly employed in a number of hydrometeorological applications (Chau, 2017; Fotovatikhah et al., 2018; Taormina et al., 2015; Wang et al., 2013)."

Some inconsistencies and minor errors that needed attention are: ïAËŻnËĞ Replace ": : :had a slight better: : :" with ": : :had a slightly better: : :" in line 250 of p.13 22. In the conclusion section, the limitations of this study and suggested improvements of this work should be highlighted. Reply: The authors appreciate this reviewer's detailed comment. The suggested minor error was corrected accordingly, and the conclusion was modified accordingly as the following.

[revised manuscript text omitted]

1977). Compared to the MC model requiring a significant number of parameters to generate multisite occurrence, the multisite occurrence model proposed by Wilks (1998) transforms the standard normal variate and simulates the sequence with multivariate normal distribution, and then back-transforms the multivariate normal sequence to the original domain. The model is able to reproduce the contemporaneous multisite dependence structure and lagged dependence only for the same site while requiring a complex simulation process to estimate parameter for each site and being unable to preserve lagged dependence between sites.

[revised manuscript text omitted]

---

## Referee Comment (RC2) · Anonymous Referee #2 · 31 Dec 2018

Present study attempts to develop a novel simulation method for multi-site precipitation occurrence, combining the k-nearest neighbor sampling technique and genetic algorithm. The coupled model has been applied in precipitation occurrence simulation in single sites. The (only) novelty probably lies in the application of this coupled technique in generating the multi-site precipitation occurrence. Authors may clarify these and may specify whether the novelty lies in the method deployed or in the application (See line 35 in the abstract and further such claims in the manuscript body). While, stochastic weather models (like the one deployed in this study) are commonly deployed in various applications, it would be preferable to give some physical justification to the application and comprehend the results obtained. This would bring more confidence

into the purely statistical methods which otherwise may not have captured any physical relationships/behavior of the system been dealt. This is particularly significant in the present study, since multi-site occurrences might be directed by many climatic feedbacks and also controlled by many local factors also. Absence of any such physical explanation may leave the methods sound robotic and put doubts in its generic applicability. In addition, the present method is compared with a method (MONR) which is developed almost two decades back. Is MONR a frequently used method for multi-site precipitation occurrence simulation? It would be convincing to compare the present technique with more recent methods deployed for multi-site precipitation occurrence simulation. More specific comments are provided below for the kind consideration of the authors.

1. Line 68 – 74: Wilks (1998) model assumes standard normal variate and underestimates the lagged cross correlation. As mentioned before, is it really worth to compare the present method to this model, which works on an entirely different hypothesis? As mentioned by the authors in the next paragraph (lines 75-81), KNNR and KNNR-GA are proved to be efficient. Won't it be better to compare the present model (DKNNR) to compare with the above model, to highlight its applicability in multi-site precipitation occurrence, given that the novelty of the study is claimed to be in this application.

2. Line 78-81: It is mentioned that KNNR model cannot produce different patterns and coupling with GA solves this drawback. Please provide more details on how GA could possibly solve this. And how the application of GA could ensure generation of similar populations. It would be interesting if some physical sense can also be provided here – how possibly GA could simulate those system behavior?

3. Line 142: "multisite occurrence X and the observed multisite occurrence x". Aren't both these variables multi-dimensional and of same size? It would be ideal to denote both in capitals then.

4. Line 158: When the algorithm will select the GA mixing? What is the criterion for

GA mixing in the procedure?

5. Line 178-179: It is mentioned later in the manuscript that the changes in the mutation and cross-over probabilities may be carried out to adapt to the changes in the transition and marginal probability distributions (See lines 187-188). Considering that, would it be ideal to fix these as 0.01, following Lee et al (2010b). Shouldn't this be case specific? If not then, the later statement (lines 187-188) are questionable.

6. Section 3.2: Authors must be pointing towards "Dealing with Non-stationarity" than "Adaptation to climate change". It is clear that only changes in marginal and transition probabilities are been considered, by tuning the crossover and mutation probabilities? "Climate change" may refer to a larger phenomenon, which might not be addressed directly in the present study. Please explain.

7. How tuning of crossover and mutation probabilities could handle the non-stationarity in the time series of multiple stations? Can the model change these parameters in between the time frame of the simulation, so as to incorporate the parameter change(s) in the probability distributions?

8. Section 4: Please provide more details about the precipitation data used, its seasonality, rainy day characteristics etc. Are the stations selected meteorologically homogenous?

9. Section 5: This may go into the results section, if it sounds fine.

10. Line 222: " ….., since a synoptic scale weather system could result in lagged cross-correlation" – Can this statement be generalized for all locations?

11. Figure 2-4: Ensemble means from MONR are close to the observed mean, than those of DKNNR model. Is MONR better in that sense? Please clarify.

12. Line 254-255: "Even though the transition probabilities were not employed in simulating rainfall occurrence, the DKNNR model preserved this statistic fairly well" – Is it merely by chance? Please provide justification to build confidence. Do you expect the

results to vary, when deployed in different regions?

13. Line 273-274: "Precipitation is not significantly correlated with more than one day" – Please provide reference. The statement may not hold well globally, as Box-Jenkins models of higher order are often applied for simulating precipitation events.

14. It would be better to number the stations considering its proximity. It will help in analyzing the results.

15. It would be interesting to see the results generated by the simple KNNR model in this application. Also, it would be helpful, if you may please explain how the incorporation of GA possibly helped in modeling the physical laws of the precipitation system.

16. Disadvantage of the simple KNNR model is the inability to simulate different patterns from the observed series. Do the stations selected exhibit significant non-stationarity? If not, will the KNNR model also serve the purpose?

17. Section 6.3: I am a little confused here. How can the parameters be changed in the future, for the model to adapt to the future changes, given that we may not clear information about these changes?

---

## Author Comment (AC2) · 13 Jan 2019

*Reply: The authors appreciate this reviewer's comments. The authors have improved the quality of the current study according to the comments of the reviewer. Hope this reviewer is satisfied with this modification.*

Present study attempts to develop a novel simulation method for multi-site precipitation occurrence, combining the k-nearest neighbor sampling technique and genetic algorithm. The coupled model has been applied in precipitation occurrence simulation in single sites. The (only) novelty probably lies in the application of this coupled technique in generating the multi-site precipitation occurrence. Authors may clarify these and may specify whether the novelty lies in the method deployed or in the application (See line 35 in the abstract and further such claims in the manuscript body).

*Reply: The authors appreciate this reviewer's insightful comment. The novelty of the current study is to propose the discrete version of KNNR-GA model in simulating multisite occurrence. The KNNR-GA model has been developed for multisite simulation of streamflow for continuous variables. The novelty of the current study is how to handle the multisite discrete binary process which is the main difference between the continuous version and the discrete version of the current study. The authors have improved the abstract and manuscript to emphasize this point. Hope this modification is satisfactory.*

While, stochastic weather models (like the one deployed in this study) are commonly deployed in various applications, it would be preferable to give some physical justification to the application and comprehend the results obtained. This would bring more confidence into the purely statistical methods which otherwise may not have captured any physical relationships/behavior of the system been dealt. This is particularly significant in the present study, since multi-site occurrences might be directed by many climatic feedbacks and also controlled by many local factors also. Absence of any such physical explanation may leave the methods sound robotic and put doubt s in its generic applicability.

*Reply: The authors have tried to provide the physical connection to the current results. For example, the following statement for the GA mixing process has been connected with the physical process of the proposed model.*

*"This can be problematic for the simulation purpose in that one of the major simulation purposes is to simulate sequences that might possibly happen in future. The wet (1) or dry (0)*

*for multisite precipitation occurrence is decided by the spatial distribution of a precipitation weather system. A humid air mass can be distributed randomly relying on wind velocity and direction as well as surrounding air pressure. In general, any combinations of wet and dry stations can be possible, especially when the simulation continues infinitely. Therefore, the patterns of simulated data must be allowed to have any possible combinations, here 4096 even if it has not been observed from the historical records. Also, its probability to have this new pattern must not be high since it has not been observed in the historical records and this can be taken into account by low probability of the crossover and mutation. "*

*"Daily precipitation occurrence, in general, shows the strongest serial correlation at lag-1 and its correlation decays as the lag gets longer. This is because a precipitation weather system moves according to the surrounding pressure and wind direction that dynamically change within a day or week. Therefore, we analyzed the lag-1 cross-correlation in the current study as the representative lagged correlation structure."*

*"In the DKNNR modeling procedure, the simple distance measurement in Eq. (11) allows to preserve transition probabilities in that the following multisite occurrence is resampled from the historical data whose previous states of multisite occurrence ($x_i^s$) are similar to the current simulation multisite occurrence ($X_c^s$). This summarized distance ($D_i$) is an essential tool in the proposed DKNNR modeling. The condition of the current weather system is memorized and the system is conditioned on simulating the following multisite occurrence with the distance measurement like a precipitation weather system dynamically changes but often it impacts the system of the following day."*

In addition, the present method is compared with a method (MONR) which is developed almost two decades back. Is MONR a frequently used method for multi-site precipitation occurrence simulation? It would be convincing to compare the present technique with more recent methods deployed for multi-site precipitation occurrence simulation. More specific comments are provided below for the kind consideration of the authors.

*Reply: The authors appreciate the reviewer's insightful comment. Even if MNOR model is rather old-fashioned, this model has been popularly employed in this field and its performance is more comparable to the Markov Chain model especially in multisite occurrence cases of precipitation dataset.*

1. Line 68 – 74: Wilks (1998) model assumes standard normal variate and underestimates the lagged cross correlation. As mentioned before, is it really worth to compare the present method to this model, which works on an entirely different hypothesis? As mentioned by the authors in the next paragraph (lines 75-81), KNNR and KNNR-GA are proved to be efficient. Won't it be better to compare the present model (DKNNR) to compare with the above model, to highlight its applicability in multi-site precipitation occurrence, given that the novelty of the study is claimed to be in this application.

*Reply: The authors appreciate the reviewer's insightful comment. The MONR model is the model of Wilks (1998) and it has been popularly employed in the literature. The present study compared the discrete version of KNNR-GA with the model of Wilks (1998), named as MONR here. See the first line of the section 2.2 as the following:*

*"Wilks (1998) suggested a multisite occurrence model using a standard normal random number (here, denoted as MONR) that is spatially dependent but serially independent."*

2. Line 78-81: It is mentioned that KNNR model cannot produce different patterns and coupling with GA solves this drawback. Please provide more details on how GA could possibly solve this. And how the application of GA could ensure generation of similar populations. It would be interesting if some physical sense can also be provided here – how possibly GA could simulate those system behavior?

*Reply: The authors appreciate the reviewer's detailed comment. Further explanation is added in the manuscript to improve the clarity in the result section.*

*"We further tested and discuss why the GA mixing is necessary in the proposed DKNNR model as follows. For example, assume that three weather stations are considered and observed data only has the occurrence cases of 000, 001,011,010, 011,100,111 among $2^3$=8 possible cases. In other words, no patterns for 110 and 101 is found in the observed data. Note that 0 is dry day and 1 is rainy (or wet) day. The KNNR is a resampling process in that the simulation data is resampled from the observation. Therefore, no new patterns such as 110 and 101 can be found in the simulated data.*
*This can be problematic for the simulation purpose in that one of the major simulation purposes is to simulate sequences that might possibly happen in future. The wet (1) or dry (0) for multisite precipitation occurrence is decided by the spatial distribution of a precipitation weather system. A humid air mass can be distributed randomly relying on wind velocity and direction as well as surrounding air pressure. In general, any combinations of wet and dry stations can be possible, especially when the simulation continues infinitely. Therefore, the patterns of simulated data must be allowed to have any possible combinations, here 4096 even if it has not been observed from the historical records. Also, its probability to have this new pattern must not be high since it has not been observed in the historical records and this can be taken into account by low probability of the crossover and mutation.*
*This drawback of the KNNR model frequently happens in multisite occurrence as the number of stations increases. Note that the number of patterns increases as $2^n$ where n is the number of stations. If n=12, then 4096 cases must be observed. However, among 4096 cases, observed cases are limited, since the number of data is limited. The GA process can mix two candidate patterns to produce new patterns. For example, in the three station case, a new pattern 101 can be produced from two observed occurrence candidates of 001 and 100 by the crossover of the first value of 001 to the first value of 100 (i.e. 001 →101), which is not in the observed data.*
*Note that the data employed in the case study are 40 years and 122 days (summer months) in each year. The total number of the observed data is 4880 and the number of possible cases is 4096. We checked how many of possible cases are not found in the observed data. The result shows that 3379 cases are not observed at all for the entire cases as shown in Figure 4.*
*We further investigated how many new patterns are generated with the probabilities $P_{cr}$=0.02, $P_m$=0.001 by the proposed GA mixing. The generated data for 100 sequences from DKNNR with the GA mixing shows that the number 3379 was reduced to 1200, which is not in the dataset among the 4096 possible patterns. Therefore, more than 2000 new patterns were simulated with the GA mixing process. The KNNR model without the GA mixing does not produce any new patterns in the 100 sequences with the same length of the historical data."*

[Figure]

Figure S 1. Frequency of the observed patterns among all the possible cases (4096). The X coordinate indicates each pattern. All zero (0) and all one (4095) has the largest and second largest number of frequency (i.e. 1894 and 877, respectively) as expected meaning all dry and all wet stations. Note that the bars are very sporadic indicating a number of occurrence patterns are not observed.

3. Line 142: "multisite occurrence X and the observed multisite occurrence x". Aren't both these variables multi-dimensional and of same size? It would be ideal to denote both in capitals then.

*Reply: The authors appreciate the reviewer's detailed comment. We denote the observed occurrence with a lower case and the simulate variable with an upper case. For representing*

*a multisite variable, we use the bold character. This separation is inevitable to express the simulation procedure from the observed dataset (especially in KNNR model). In Eq.11, $X_c^s$ and $x_i^s$ represent only the simulation variable and observed data of the $s^{th}$ station. Hope this is reasonable to this reviewer. To avoid confusion, we modify the sentence as follows:*

*"Estimate the distance between the current (i.e. time index: c) multisite occurrence $X_c^s$ and the observed multisite occurrence $x_i^s$ for the $s^{th}$ station s=1,...,S. Here, the distance is measured for i=1,..., n-1 as*

$$D_i = \sum_{s=1}^{S} \left| X_c^s - x_i^s \right| \qquad (1)$$

*"*

4. Line 158: When the algorithm will select the GA mixing? What is the criterion for GA mixing in the procedure?

*Reply: The authors appreciate the reviewer's insightful comment. It is subjective. If one wants to simulate the dataset as the same observed pattern, this procedure can be skipped. Otherwise, the GA procedure gives the benefit of generating new patterns that we already discussed under comment 2. The sentence is modified accordingly.*
*"Execute the following steps for GA mixing if GA mixing is subjectively selected. Otherwise, skip this step."*

5. Line 178-179: It is mentioned later in the manuscript that the changes in the mutation and cross-over probabilities may be carried out to adapt to the changes in the transition and marginal probability distributions (See lines 187-188). Considering that, would it be ideal to fix these as 0.01, following Lee et al (2010b). Shouldn't this be case specific? If not then, the later statement (lines 187-188) are questionable.

*Reply: From the comment of the Reviewer 1, the estimation of parameter set was reinvestigated thoroughly. We concluded that the parameter set of $P_{cr}$ and $P_m$ as 0.02 and 0.003 showed the best from the result of RMSE estimated with the transition and limiting probabilities of the tested stations. The detailed results are as follows. Hope this investigation is satisfactory.*

*"The roles of crossover probability $P_{cr}$ (Eq. (13)) and mutation probability $P_m$ (Eq.(14)) were studied by Lee et al. (2010b). In the current study, we further tested by selecting an appropriate parameter set of these two parameters with the simulated data from the DKNNR model and the record length of 100,000. RMSE (Eq. (18)) of the three transition and limiting probabilities ($P_{11}$, $P_{01}$, and $P_1$) between the simulated data and the observed was used, since those probabilities are key statistics that the simulated data must match with the observed data and no parameterization of these probabilities was made for the current DKNNR model. Results are shown in Figure 2 and Figure 3 for $P_{cr}$ and $P_m$, respectively. For $P_{cr}$ in Figure 2, the probability of 0.02 shows the smallest RMSE in all transition and limiting probabilities. The RMSE of $P_m$ in Figure 3 shows a slight fluctuation along with $P_m$. However, all three probabilities ($P_{11}$, $P_{01}$, and $P_1$) have relatively small RMSEs in $P_m$ =0.003. Therefore, the parameter set 0.02 and 0.003 was chosen for $P_{cr}$ and $P_m$, respectively, and employed in the current study."*

[Figure]

*Figure 2. Testing for different probabilities of crossover Pcr. RMSE is estimated for all the tested 12 stations for each transition probability.*

[Figure]

*Figure 3. Testing for different probabilities of mutation Pm. RMSE is estimated for all the tested 12 stations for each transition probability.*

6. Section 3.2: Authors must be pointing towards "Dealing with Non-stationarity" than "Adaptation to climate change". It is clear that only changes in marginal and transition probabilities are been considered, by tuning the crossover and mutation probabilities? "Climate change" may refer to a larger phenomenon, which might not be addressed directly in the present study. Please explain.

*Reply: The authors totally agree with the concern of the reviewer. Tuning the crossover and mutation probabilities only affected the marginal and transition probabilities. This limitation must be addressed as this reviewer commented. We added the following to address the*

*concern from this reviewer at the end of section 6. The authors hope that this statement is satisfactory.*
*"Climate change, however, may refer to a larger phenomenon, which cannot be addressed directly through modifying only the marginal and transition probabilities as in the current study. Further modeling development on systematically varying temporal and spatial cross-correlations is required to properly address the climate change of the regional precipitation system."*

7. How tuning of crossover and mutation probabilities could handle the non-stationarity in the time series of multiple stations? Can the model change these parameters in between the time frame of the simulation, so as to incorporate the parameter change(s) in the probability distributions?

*Reply: The authors totally agree with the concern of the reviewer as with the previous comment that tuning the crossover and mutation probabilities only effected the marginal and transition probabilities. The authors consider that it is possible that the model can change the parameter to adapt to the climate change between the time frame of the simulation to incorporate the parameter change automatically. But this capability has not been fully investigated. In addition, the focus of the current study is to propose a novel approach that simulates multisite occurrence process through the nonparametric approaches. Further development for adopting to climate change and its application is presented as a possible improvement of the proposed model in the near future and will be presented as a separate work as explained in the conclusion section as the following.*

*"We tested further the enhancement of the proposed model for adapting to climate change by modifying the mutation and crossover probabilities $P_m$ and $P_{cr}$. The results showed that the proposed DKNNR model has the capability to adapt to the climate change scenarios, but further elaborate work is required to find the best probability estimation for climate change. Also, only the marginal and transition probabilities cannot address the climate change of regional precipitation. The variation of temporal and spatial cross-correlation structure must be considered to properly address the climate change of the regional precipitation system. Further study on improving the model adaptability to climate change will be followed in the near future.Also, the simulated multisite occurrence can be coupled with a multisite amount model to produce precipitation events, including zero values. Further development can be made for multisite amount models with a nonparametric technique, such as KNNR and bootstrapping."*

8. Section 4: Please provide more details about the precipitation data used, its seasonality, rainy day characteristics etc. Are the stations selected meteorologically homogenous?

*Reply: The authors appreciate the reviewer's detailed comment. The following is added to address this comment. Hope this statement is satisfactory.*

*"The employed precipitation dataset presents strong seasonality, since this area is dry from late fall to early autumn and humid and rainy during the remaining seasons, especially in summer. The employed stations are not far from each other, at most 100 km apart, and not much high mountains are located in the current study area. Therefore, this region can be considered as a homogeneous region (Lee et al., 2007)."*

*"To validate the proposed model appropriately, test sites must be highly correlated with each other as well as have significant temporal relation. The stations inside the Yeongnam area cover one of the most important watersheds, the Nakdong River basin, where the Nakdong River passes through the entire basin and its hydrological assessments for agriculture and climate change have a particular value in flood control and water resources management such as floods and droughts."*

9. Section 5: This may go into the results section, if it sounds fine.

*Reply: The authors appreciate the reviewer's comment. The authors separate this section to explain how the developed model is applied to the datasets and what measurements were used to show its performance. The authors consider that the separation of this application part is reasonable because there are no specific results in this section. The results of the GA mixing and its probability section in the result section are also added for the comments of the reviewer.*

10. Line 222: " . . ..., since a synoptic scale weather system could result in lagged cross-correlation" – Can this statement be generalized for all locations?

*Reply: The authors appreciate the reviewer's specific comment and understand his concern. The statement might not be always true. Therefore, the sentence was modified accordingly as follows:*

 *"In the current study, this statistic was analyzed, since a synoptic scale weather system often results in lagged cross-correlation for daily precipitation data (Wilks, 1998)."*

11. Figure 2-4: Ensemble means from MONR are close to the observed mean, than those of DKNNR model. Is MONR better in that sense? Please clarify.

*Reply: The authors agree with the reviewer's comment and it is already mentioned in the manuscript as the following (see the L250-251). We also modified the sentence to include the same implication to P01 and P1 as well as P11.*
*"It seems that the MONR model had a slightly better performance since this statistic is parameterized in the model as shown in section 2.2 and that is the same for P01 and P1 as shown in Figure 5 and Figure 6."*

12. Line 254-255: "Even though the transition probabilities were not employed in simulating rainfall occurrence, the DKNNR model preserved this statistic fairly well" – Is it merely by chance? Please provide justification to build confidence. Do you expect the results to vary, when deployed in different regions?

*Reply: The authors appreciate the reviewer's crucial comment. The KNN resampling with the distance in Eq. (11) between the current simulation multisite occurrence ($X_c^s$ ) and the historical multisite occurrence states ($x_i^s$) allows to preserve the transition probabilities. The following statement is added accordingly.*

 *"In the DKNNR modeling procedure, the simple distance measurement in Eq. (1) allows to preserve transition probabilities in that the following multisite occurrence is resampled from the historical data whose previous states of multisite occurrence ($x_i^s$) are similar to the*

*current simulation multisite occurrence ($X_c^s$). This summarized distance ($D_i$) is an essential tool in the proposed DKNNR modeling. The condition of the current weather system is memorized and the system is conditioned on simulating the following multisite occurrence with the distance measurement like a precipitation weather system dynamically changes but often it impacts the system of the following day.”*

13. Line 273-274: “Precipitation is not significantly correlated with more than one day” – Please provide reference. The statement may not hold well globally, as Box-Jenkins models of higher order are often applied for simulating precipitation events.

*Reply: The authors totally agree with the reviewer's comment. The sentence was modified accordingly. Hope this modification is satisfactory.*

*“Daily precipitation occurrence, in general, shows the strongest serial correlation at lag-1 and its correlation decays as the lag gets longer. This is because a precipitation weather system moves according to the surrounding pressure and wind direction that dynamically change within a day or week. Therefore, we analyzed the lag-1 cross-correlation in the current study as the representative lagged correlation structure.”*

14. It would be better to number the stations considering its proximity. It will help in analyzing the results.

*Reply: The authors appreciate the reviewer's comment. The author tried to change the numbers but consider that this may not be meaningful much since the order from west to east or north to south can be different with its numbering. Readers might be confused from this numbering. For example, the current 8,7,6, 10,2,9,1 stations can be changed to 1,2,3,4,5,6,7. The stations 3 and 4 seem close to each other due to renumbering, which is not correct. We also tested with 1,2,3,7,6,5,4. However, 1 and 7 must be far away from each other according to its numbering but they are very close to each other. We tried different numbering to consider the proximity but did not find any logical ordering. Therefore, we prefer staying as it is. Hope this can be understandable to the reviewer.*

15. It would be interesting to see the results generated by the simple KNNR model in this application. Also, it would be helpful, if you may please explain how the incorporation of GA possibly helped in modeling the physical laws of the precipitation system.

*Reply: The authors appreciate the reviewer's insightful comment. We produced the results without the GA process as presented in the following (See Figure S2-Figure S6). The presented results show that no significant difference from the one with the GA mixing can be found. The following is discussed in the manuscript right before the results of the probability selection (section 6.1).*

*“We also tested the simulation without the GA mixing procedure (results not shown). The results showed that no better result could be found from the simulation without GA mixing. The necessity of the GA mixing is further discussed in the following.”*

[Figure]

Figure S 2. Boxplots of the P11 probability for the data simulated from the DKNNR model without the GA mixing (top panel) and the MONR model (bottom panel) as well as the observed (x marker) for the 12 selected weather stations from the Yeongnam province.

[Figure]

Figure S 3. Boxplots of the P01 probability for the data simulated from the DKNNR model without the GA mixing (top panel) and the MONR model (bottom panel) as well as the observed (x marker) for the 12 selected weather stations from the Yeongnam province.

[Figure]

Figure S 4. Boxplots of the P1 probability for the data simulated from the DKNNR model without the GA mixing (top panel) and the MONR model (bottom panel) as well as the observed (x marker) for the 12 selected weather stations from the Yeongnam province.

[Figure]

[Figure]

Figure S 5. Scatterplot of cross-correlations between 12 weather stations for the observed data (X coordinate) and the generated data (Y coordinate) generated from the DKNNR model without the GA mixing (top panel) and the MONR model (bottom panel). The cross-correlations from 100 generated series are averaged for the filled circle and the errorbars upper and lower extended lines indicate the range of 1.95×standard deviation.

[Figure]

[Figure]

Figure S 6. Scatterplot of lag-1 cross-correlations between 12 weather stations for the observed data (X coordinate) and the generated data (Y coordinate) generated from the DKNNR model without the GA mixing (top panel) and the MONR model (bottom panel). The cross-correlations from 100 generated series are averaged for the filled circle and the errorbars upper and lower extended lines indicate the range of 1.95×standard deviation

16. Disadvantage of the simple KNNR model is the inability to simulate different patterns from the observed series. Do the stations selected exhibit significant nonstationarity? If not, will the KNNR model also serve the purpose?

*Reply: The authors appreciate the reviewer's comment. The GA mixing was not applied for nonstationarity. The GA mixing is applied to overcome the disadvantage of the KNNR model that only observed pattern is repeated in the simulated data. This case is not sound for the simulation study purpose. As mentioned under comment 2, more than half of the possible patterns are not observed in the historical data. This has been covered multiple times already under previous comments. Hope this explanation can be acceptable to the reviewer.*

17. Section 6.3: I am a little confused here. How can the parameters be changed in the future, for the model to adapt to the future changes, given that we may not clear information about these changes?

*Reply: The authors appreciate the reviewer's comment. The authors did not fully investigate the specific changes required to be made for specific climate change assessment at this stage. As mentioned under comment 7, the focus of the current study is to propose a novel approach that simulates multisite occurrence process through nonparametric approaches. Further development for adopting to climate change and its application are partially presented as a possible improvement of the proposed model in the near future and will be presented as a separate work as explained in the conclusion. This limitation and possible development are discussed in the last section.*

---

## Referee Report (RR1)

**Comments on "Discrete k-nearest neighbor resampling for simulating multisite precipitation occurrence and adaption to climate change" by Taesam Lee and Vijay P. Singh**

Authors have addressed some of the comments satisfactorily. However, clarifications are needed on a few responses. I am highlighting those below.

**1.**

Present study attempts to develop a novel simulation method for multi-site precipitation occurrence, combining the k-nearest neighbor sampling technique and genetic algorithm. The coupled model has been applied in precipitation occurrence simulation in single sites. The (only) novelty probably lies in the application of this coupled technique in generating the multi-site precipitation occurrence. Authors may clarify these and may specify whether the novelty lies in the method deployed or in the application (See line 35 in the abstract and further such claims in the manuscript body).

*Reply: The authors appreciate this reviewer's insightful comment. The novelty of the current study is to propose the discrete version of KNNR-GA model in simulating multisite occurrence. The KNNR-GA model has been developed for multisite simulation of streamflow for continuous variables. The novelty of the current study is how to handle the multisite discrete binary process which is the main difference between the continuous version and the discrete version of the current study. The authors have improved the abstract and manuscript to emphasize this point. Hope this modification is satisfactory.*

In response to the general comment of highlighting the novelty of the work, modified abstract says "Multisite occurrence model with standard normal variate (MONR) has been used preserving key statistics and contemporaneous correlation in literature, but it cannot reproduce lagged crosscorrelation between stations and long stochastic simulation is required to estimate its parameters. Employing a nonparametric technique, k-nearest neighbor resampling (KNNR), and coupling it with Genetic Algorithm (GA), this study proposes a novel simulation method for multisite precipitation occurrence overcoming the shortcomings of the existing MONR model." **This sounds as if the focus of the study itself is only to overcome the limitations of MONR model. The novelty (if any) is still not brought out clearly.**

**2.**

In addition, the present method is compared with a method (MONR) which is developed almost two decades back. Is MONR a frequently used method for multi-site precipitation occurrence simulation? It would be convincing to compare the present technique with more recent methods deployed for multi-site precipitation occurrence simulation. More specific comments are provided below for the kind consideration of the authors.

*Reply: The authors appreciate the reviewer's insightful comment. Even if MNOR model is rather old-fashioned, this model has been popularly employed in this field and its performance is more comparable to the Markov Chain model especially in multisite occurrence cases of precipitation dataset.*

**A few recent studies are given below on the same topic, which focus on the same topic – multi-site precipitation occurrence.**
Evin et al., HESS, 2018: Stochastic generation of multi-site daily precipitation focusing on extreme events

Mehrotra et al., JH, 2006: A comparison of three stochastic multi-site precipitation occurrence generators

**3.**

> 1. Line 68 – 74: Wilks (1998) model assumes standard normal variate and underestimates the lagged cross correlation. As mentioned before, is it really worth to compare the present method to this model, which works on an entirely different hypothesis? As mentioned by the authors in the next paragraph (lines 75-81), KNNR and KNNR-GA are proved to be efficient. Won't it be better to compare the present model (DKNNR) to compare with the above model, to highlight its applicability in multi-site precipitation occurrence, given that the novelty of the study is claimed to be in this application.
>
> *Reply: The authors appreciate the reviewer's insightful comment. The MONR model is the model of Wilks (1998) and it has been popularly employed in the literature. The present study compared the discrete version of KNNR-GA with the model of Wilks (1998), named as MONR here. See the first line of the section 2.2 as the following:*
> *"Wilks (1998) suggested a multisite occurrence model using a standard normal random number (here, denoted as MONR) that is spatially dependent but serially independent."*

**Please clarify how the results would be different for DKNNR and KNNR models?**

**4.**

> 2. Line 78-81: It is mentioned that KNNR model cannot produce different patterns and coupling with GA solves this drawback. Please provide more details on how GA could possibly solve this. And how the application of GA could ensure generation of similar populations. It would be interesting if some physical sense can also be provided here – how possibly GA could simulate those system behavior?
>
> *Reply: The authors appreciate the reviewer's detailed comment. Further explanation is added in the manuscript to improve the clarity in the result section.*

**The authors have explained the need for GA in the methodology, to simulate the patterns different from the historical patterns. This is understood. However, it is not clear how GA will be trained to generate those patterns specific to the study area. I am sure that GA might generate many unwanted patterns also, which is not physically possible in the study region. How GA is supposed to avoid this unwanted patterns?**

**5.**

> 4. Line 158: When the algorithm will select the GA mixing? What is the criterion for GA mixing in the procedure?
>
> *Reply: The authors appreciate the reviewer's insightful comment. It is subjective. If one wants to simulate the dataset as the same observed pattern, this procedure can be skipped. Otherwise, the GA procedure gives the benefit of generating new patterns that we already discussed under comment 2. The sentence is modified accordingly.*
> *"Execute the following steps for GA mixing if GA mixing is subjectively selected. Otherwise, skip this step."*

**So, is it up on the user to opt for GA mixing? It should have been based on the properties of the time series and study region. If the rainfall exhibits more or less an unchanging pattern across the stations, then the future pattern can be found in the historical patterns too. In that case GA mixing could be avoided. The algorithm should have the criterion for that.**

**6.**

> 6. Section 3.2: Authors must be pointing towards "Dealing with Non-stationarity" than "Adaptation to climate change". It is clear that only changes in marginal and transition probabilities are been considered, by tuning the crossover and mutation probabilities? "Climate change" may refer to a larger phenomenon, which might not be addressed directly in the present study. Please explain.
>
> *Reply: The authors totally agree with the concern of the reviewer. Tuning the crossover and mutation probabilities only affected the marginal and transition probabilities. This limitation must be addressed as this reviewer commented. We added the following to address the*

**Thanks for agreeing to this comment. In that case, there is an over-emphasis in the title regarding the "adaptation to climate change". If the methodology is not addressing the climate change, please remove the section or modify it accordingly. Section 5.4 still claims "Adaptation to climate change". This can be addressed along with the next comment (7th comment), where again authors justify the changing of these probabilities to address the climate change. It is not clear, how tuning of crossover and mutation probabilities could handle the non-stationarity (or climate change according to authors) in the time series of multiple stations?**

**7.**

> 15. It would be interesting to see the results generated by the simple KNNR model in this application. Also, it would be helpful, if you may please explain how the incorporation of GA possibly helped in modeling the physical laws of the precipitation system.
>
> *Reply: The authors appreciate the reviewer's insightful comment. We produced the results without the GA process as presented in the following (See Figure S2-Figure S6). The presented results show that no significant difference from the one with the GA mixing can be found. The following is discussed in the manuscript right before the results of the probability selection (section 6.1).*

**I could not find much difference between simple KNNR model and KNNR model with GA mixing (Figures s2-s6 and Figures 5-9). Both produce almost same results. Does that mean, the incorporation of GA has not added much value?**

**8.**

> 17. Section 6.3: I am a little confused here. How can the parameters be changed in the future, for the model to adapt to the future changes, given that we may not clear information about these changes?
>
> *Reply: The authors appreciate the reviewer's comment. The authors did not fully investigate the specific changes required to be made for specific climate change assessment at this stage. As mentioned under comment 7, the focus of the current study is to propose a novel approach that simulates multisite occurrence process through nonparametric approaches. Further development for adopting to climate change and its application are partially presented as a possible improvement of the proposed model in the near future and will be presented as a separate work as explained in the conclusion. This limitation and possible development are discussed in the last section.*

**Please see comment 6 in this document, regarding the adaptation to climate change.**

Kind Regards.

---

## Author Response (AR2)

**Comments on "Discrete k-nearest neighbor resampling for simulating multisite precipitation occurrence and adaption to climate change" by Taesam Lee and Vijay P. Singh**

Authors have addressed some of the comments satisfactorily. However, clarifications are needed on a few responses. I am highlighting those below.

**1.**

Present study attempts to develop a novel simulation method for multi-site precipitation occurrence, combining the k-nearest neighbor sampling technique and genetic algorithm. The coupled model has been applied in precipitation occurrence simulation in single sites. The (only) novelty probably lies in the application of this coupled technique in generating the multi-site precipitation occurrence. Authors may clarify these and may specify whether the novelty lies in the method deployed or in the application (See line 35 in the abstract and further such claims in the manuscript body).

*Reply: The authors appreciate this reviewer's insightful comment. The novelty of the current study is to propose the discrete version of KNNR-GA model in simulating multisite occurrence. The KNNR-GA model has been developed for multisite simulation of streamflow for continuous variables. The novelty of the current study is how to handle the multisite discrete binary process which is the main difference between the continuous version and the discrete version of the current study. The authors have improved the abstract and manuscript to emphasize this point. Hope this modification is satisfactory.*

In response to the general comment of highlighting the novelty of the work, modified abstract says "Multisite occurrence model with standard normal variate (MONR) has been used preserving key statistics and contemporaneous correlation in literature, but it cannot reproduce lagged crosscorrelation between stations and long stochastic simulation is required to estimate its parameters. Employing a nonparametric technique, k-nearest neighbor resampling (KNNR), and coupling it with Genetic Algorithm (GA), this study proposes a novel simulation method for multisite precipitation occurrence overcoming the shortcomings of the existing MONR model." **This sounds as if the focus of the study itself is only to overcome the limitations of MONR model. The novelty (if any) is still not brought out clearly.**

Reply: The authors appreciate the comment. The abstract was modified to bring out the novelty of the current study accordingly. Hope this modification satisfactory.

Stochastic weather simulation models are commonly employed in water resources management, agricultural applications, forest management, transportation management, and recreational activities. Stochastic simulation of multisite precipitation occurrence is a challenge because of its intermittent characteristics as well as spatial and temporal cross-correlation. This study proposes a novel simulation method for multisite precipitation occurrence employing a nonparametric technique, the discrete version of the k-nearest neighbor resampling (KNNR), and coupling it with Genetic Algorithm (GA). Its modification for the study of climatic change adaptation is also tested. The datasets simulated from both the DKNNR model and an existing traditional model were evaluated using a number of statistics, such as occurrence and transition probabilities as well as temporal and spatial cross-correlations. Results showed that the proposed DKNNR model with GA simulated multisite precipitation occurrence preserved the lagged crosscorrelation between sites while the existing conventional model was not able to reproduce lagged crosscorrelation between stations, so long stochastic simulation was required.  Also, the GA mixing process provided a number of new patterns that were different from observations, which was not feasible with the sole DKNNR model.  When climate change was considered, the model performed satisfactorily, but further improvement is required to more accurately simulate specific variations of the occurrence probability.

**2.**

In addition, the present method is compared with a method (MONR) which is developed almost two decades back. Is MONR a frequently used method for multi-site precipitation occurrence simulation? It would be convincing to compare the present technique with more recent methods deployed for multi-site precipitation occurrence simulation. More specific comments are provided below for the kind consideration of the authors.

*Reply: The authors appreciate the reviewer's insightful comment. Even if MNOR model is rather old-fashioned, this model has been popularly employed in this field and its performance is more comparable to the Markov Chain model especially in multisite occurrence cases of precipitation dataset.*

**A few recent studies are given below on the same topic, which focus on the same topic – multi-site precipitation occurrence.**

Evin et al., HESS, 2018: Stochastic generation of multi-site daily precipitation focusing on extreme events

Mehrotra et al., JH, 2006: A comparison of three stochastic multi-site precipitation occurrence generators

Reply: The authors appreciate the comment providing highly relevant studies. The provided references were mentioned and cited in the current manuscript.

"The model is able to reproduce the contemporaneous multisite dependence structure and lagged dependence only for the same site but it requires a complex simulation process to estimate parameters for each site and is unable to preserve lagged dependence between sites. Also, a recent improvement has also been made, but the weakness of the model in Wilks (1998) was not significantly improved (Evin et al., 2018; Mehrotra et al., 2006; Srikanthan and Pegram, 2009)."

**3.**

1. Line 68 – 74: Wilks (1998) model assumes standard normal variate and underestimates the lagged cross correlation. As mentioned before, is it really worth to compare the present method to this model, which works on an entirely different hypothesis? As mentioned by the authors in the next paragraph (lines 75-81), KNNR and KNNR-GA are proved to be efficient. Won't it be better to compare the present model (DKNNR) to compare with the above model, to highlight its applicability in multi-site precipitation occurrence, given that the novelty of the study is claimed to be in this application.

*Reply: The authors appreciate the reviewer's insightful comment. The MONR model is the model of Wilks (1998) and it has been popularly employed in the literature. The present study compared the discrete version of KNNR-GA with the model of Wilks (1998), named as MONR here. See the first line of the section 2.2 as the following:*
*"Wilks (1998) suggested a multisite occurrence model using a standard normal random number (here, denoted as MONR) that is spatially dependent but serially independent."*

**Please clarify how the results would be different for DKNNR and KNNR models.**

Reply: The authors believe that KNNR model is a multivariate model, since it is dealing with multisites. However, the DKNNR can be simplified as a univariate KNNR model with range from zero to the number of stations used.  Though the result behavior might be inherited from KNNR model, its implementation is much simpler than in case of the KNNR model.

**4.**

2. Line 78-81: It is mentioned that KNNR model cannot produce different patterns and coupling with GA solves this drawback. Please provide more details on how GA could possibly solve this. And how the application of GA could ensure generation of similar populations. It would be interesting if some physical sense can also be provided here – how possibly GA could simulate those system behavior?

*Reply: The authors appreciate the reviewer's detailed comment. Further explanation is added in the manuscript to improve the clarity in the result section.*

**The authors have explained the need for GA in the methodology, to simulate the patterns different from the historical patterns. This is understood. However, it is not clear how GA will be trained to generate those patterns specific to the study area. I am sure that GA might generate many unwanted patterns also, which is not physically possible in the study region. How GA is supposed to avoid this unwanted patterns?**

Reply: The authors thank for the comment. The first procedure of the GA (genetic algorithm) is the "reproduction". The reproduction procedure is also called "the mating process" implying that one male and one female are chosen, and their genes are cross-overed and mutated to create a new offspring. In other words, the new pattern is made from the historical patterns not totally outside from the data. Therefore, the creation of unwanted patterns in the simulated data is automatically suppressed from the nature of the GA algorithm. Also, note that the mutation probability is very rare that "unwanted patterns" do not occur often. However, it happens rarely, like in nature.

The authors hope that this explanation can be acceptable to this reviewer. The manuscript is also modified accordingly to further inform readers of this issue.

"Note that the reproduction procedure of the GA allows to generate new patterns that are similar to observed patterns, but a small number of totally new patterns are simulated from the mutation procedure of the GA."

**5.**

> 4. Line 158: When the algorithm will select the GA mixing? What is the criterion for GA mixing in the procedure?
>
> *Reply: The authors appreciate the reviewer's insightful comment. It is subjective. If one wants to simulate the dataset as the same observed pattern, this procedure can be skipped. Otherwise, the GA procedure gives the benefit of generating new patterns that we already discussed under comment 2. The sentence is modified accordingly.*
> *"Execute the following steps for GA mixing if GA mixing is subjectively selected. Otherwise, skip this step."*

**So, is it up on the user to opt for GA mixing? It should have been based on the properties of the time series and study region. If the rainfall exhibits more or less an unchanging pattern across the stations, then the future pattern can be found in the historical patterns too. In that case GA mixing could be avoided. The algorithm should have the criterion for that.**

Reply: Even if the GA mixing has no criterion to choose, the GA must be applied since no one wants to simulate the patterns as the same as the historical. Of course, future pattern can be found in the historical patterns too. However, only historical patterns in the future patterns cannot be desirable, as shown in Figure 4. Hope this explanation can be acceptable to this reviewer.

**6.**

6. Section 3.2: Authors must be pointing towards "Dealing with Non-stationarity" than "Adaptation to climate change". It is clear that only changes in marginal and transition probabilities are been considered, by tuning the crossover and mutation probabilities? "Climate change" may refer to a larger phenomenon, which might not be addressed directly in the present study. Please explain.

*Reply: The authors totally agree with the concern of the reviewer. Tuning the crossover and mutation probabilities only affected the marginal and transition probabilities. This limitation must be addressed as this reviewer commented. We added the following to address the*

**Thanks for agreeing to this comment. In that case, there is an over-emphasis in the title regarding the "adaptation to climate change". If the methodology is not addressing the climate change, please remove the section or modify it accordingly. Section 5.4 still claims "Adaptation to climate change". This can be addressed along with the next comment (7th comment), where again authors justify the changing of these probabilities to address the climate change. It is not clear, how tuning of crossover and mutation probabilities could handle the non-stationarity (or climate change according to authors) in the time series of multiple stations?**

Reply: The authors consider that the major parts of the climate change adaptation studies in a stochastic generator for multisite precipitation occurrence is the capability to simulate the occurrence series with its changing probability. Even if only the marginal and transition probabilities were tested in the current study in section 3.2 and section 6.4, the current model can be further developed to handle the climate change issue. The authors believe that tuning crossover and mutation probabilities could handle this issue for each station, but not multiple stations at the same time. As mentioned in the previous reply, this tuning process cannot handle the change of correlation structure in future climate scenarios. However, the key change of marginal and transition probabilities can be adapted in the DKNNR with GA model by tuning the crossover and mutation probabilities as tested in Figure 11 and Figure 12. We agree that the tuning probabilities must be further studied to clarify whether the model works reasonably well. Also note that the crossover probabilities might affect the stations each other while the mutation probabilities do not.

To express how this tuning procedure is able to address future climate adaptation, the following description is added.
"Assume that the occurrence probability (P1) of the control period is 0.26 (see the dotted line with cross on the bottom panel of Figure 11 and Figure 12) and GCM output indicates that the occurrence probability (P1) increases up to 0.27. This can be achieved with increasing either the crossover probability to 0.1 or the mutation probability to 0.05. Note that the crossover probabilities might affect the stations each other while the mutation probabilities do not."

If this reviewer considers that this experimental part for climate change adaption is not good enough and still think this part must be removed, the authors will remove the whole part and change the title. However, the authors prefer leaving as is to indicate future development of the current model.

**7.**

15. It would be interesting to see the results generated by the simple KNNR model in this application. Also, it would be helpful, if you may please explain how the incorporation of GA possibly helped in modeling the physical laws of the precipitation system.

*Reply: The authors appreciate the reviewer's insightful comment. We produced the results without the GA process as presented in the following (See Figure S2-Figure S6). The presented results show that no significant difference from the one with the GA mixing can be found. The following is discussed in the manuscript right before the results of the probability selection (section 6.1).*

**I could not find much difference between simple KNNR model and KNNR model with GA mixing (Figures s2-s6 and Figures 5-9). Both produce almost same results. Does that mean, the incorporation of GA has not added much value?**

Reply: Note that the simple DKNNR model obtains the patterns from the historical data. Its simulated data are compared with the historical statistics. Therefore, DKNNR with GA is difficult to add much value more than the simple DKNNR except that the simulated data can have different patterns from the historical ones. However, the value of the DKNNR with GA is critical, since one of the major reasons for simulating weather data is to generate all possible cases to compare and prepare such cases.

**8.**

17. Section 6.3: I am a little confused here. How can the parameters be changed in the future, for the model to adapt to the future changes, given that we may not clear information about these changes?

*Reply: The authors appreciate the reviewer's comment. The authors did not fully investigate the specific changes required to be made for specific climate change assessment at this stage. As mentioned under comment 7, the focus of the current study is to propose a novel approach that simulates multisite occurrence process through nonparametric approaches. Further development for adopting to climate change and its application are partially presented as a possible improvement of the proposed model in the near future and will be presented as a separate work as explained in the conclusion. This limitation and possible development are discussed in the last section.*

**Please see comment 6 in this document, regarding the adaptation to climate change.**

Reply: See Reply 6.

[revised manuscript text omitted]

(6-2) Crossover: Replace each element $x^s_{p+1}$ with $x^s_{p^*+1}$ at probability $P_{cr}$, i.e.,

$$X^s_{c+1} = \begin{cases} x^s_{p^*+1} & \text{if } \varepsilon < P_{cr} \\ x^s_{p+1} & \text{otherwise} \end{cases} \qquad (13)$$

where $\varepsilon$ is a uniform random number between 0 and 1. From this crossover, a new occurrence vector whose elements are similar to the historical ones is generated.

(6-3) Mutation: Replace each element (i.e., each station, $s=1,\dots, S$) with one selected from all the observations of this element for $i=1,\dots, n$ with probability $P_m$, i.e.,

$$X^s_{c+1} = \begin{cases} x^s_{\xi+1} & \text{if } \varepsilon < P_m \\ x^s_{p+1} & \text{otherwise} \end{cases} \qquad (14)$$

[revised manuscript text omitted]

---

## Author Response (AR3)

Dear Taesam,

Thank you for you revised manuscript. I'm very pleased to report a positive response from the reviewer and I am satisfied that all substantive concerns have been addressed. I would therefore like to accept the article for publication in GMD.

The reviewer has one remaining concern relating to the reference to climate change in the title, as they believe this is overemphasised. From my perspective, I'm happy to see the climate change aspect of your work recognised in the title, however the wording "adaptation to climate change" is very broad relative to what you have done. Along with the reviewer, I would prefer something more specific to the research that makes it clear that you are adapting the simulation model to climate change scenarios (e.g. the wording used in the conclusions). I will leave the decision with you, but one option would be to write "…occurrence and model adaption to climate change".

Best Regards,
Jeff Neal

*Reply: We really appreciate all the Editors and the reviewer's for their insightful comments. Especially, our special appreciation goes to the Topical Editor, Jeff Neal.*
*For the title, the authors agree with the reviewer's concern and AE's comment. The title was modified as "Discrete k-nearest neighbor resampling for simulating multisite precipitation occurrence and model adaption to climate change" accordingly.*